# Learnable Graph Convolutional Attention Networks

**Adrián Javaloy** [1,*]   **Pablo Sánchez-Martín** [1,2,*]   **Amit Levi** [3]   **Isabel Valera** [1,4]

[1]Department of Computer Science of Saarland University, Saarbrücken, Germany
[2]Max Planck Institute for Intelligent Systems, Tübingen, Germany
[3]Huawei Noah's Ark Lab, Montreal, Canada
[4]Max Planck Institute for Software Systems, Saarbrücken, Germany

## Abstract

Existing Graph Neural Networks (GNNs) compute the message exchange between nodes by either aggregating uniformly (*convolving*) the features of all the neighboring nodes, or by applying a non-uniform score (*attending*) to the features. Recent works have shown the strengths and weaknesses of the resulting GNN architectures, respectively, GCNs and GATs. In this work, we aim at exploiting the strengths of both approaches to their full extent. To this end, we first introduce the graph convolutional attention layer (CAT), which relies on convolutions to compute the attention scores. Unfortunately, as in the case of GCNs and GATs, we show that there exists no clear winner between the three—neither theoretically nor in practice—as their performance directly depends on the nature of the data (i.e., of the graph and features). This result brings us to the main contribution of our work, the learnable graph convolutional attention network (L-CAT): a GNN architecture that automatically interpolates between GCN, GAT and CAT in each layer, by adding two scalar parameters. Our results demonstrate that L-CAT is able to efficiently combine different GNN layers along the network, outperforming competing methods in a wide range of datasets, and resulting in a more robust model that reduces the need of cross-validating.

## 1 Introduction

In recent years, Graph Neural Networks (GNNs) (Scarselli et al., 2008) have become ubiquitous in machine learning, emerging as the standard approach in many settings. For example, they have been successfully applied for tasks such as topic prediction in citation networks (Sen et al., 2008); molecule prediction (Gilmer et al., 2017); and link prediction in recommender systems (Wu et al., 2020a). These applications typically make use of message-passing GNNs (Gilmer et al., 2017), whose idea is fairly simple: in each layer, nodes are updated by aggregating the information (messages) coming from their neighboring nodes.

Depending on how this aggregation is implemented, we can define different types of GNN layers. Two important and widely adopted layers are graph convolutional networks (GCNs) (Kipf & Welling, 2017), which uniformly average the neighboring information; and graph attention networks (GATs) (Velickovic et al., 2018), which instead perform a weighted average, based on an attention score between receiver and sender nodes. More recently, a number of works have shown the strengths and limitations of both approaches from a theoretical (Fountoulakis et al., 2022; Baranwal et al., 2021; 2022), and empirical (Knyazev et al., 2019) point of view. These results show that their performance depends on the nature of the data at hand (i.e., the graph and the features), thus the standard approach is to select between GCNs and GATs via computationally demanding cross-validation.

In this work, we aim to exploit the benefits of both convolution and attention operations in the design of GNN architectures. To this end, we first introduce a novel graph convolutional attention layer (CAT), which extends existing attention layers by taking the convolved

---

*Equal contribution. Correspondence to: `{ajavaloy,sanchez}@cs.uni-saarland.de`.

features as inputs of the score function. Following (Fountoulakis et al., 2022), we rely on a contextual stochastic block model to theoretically compare GCN, GAT, and CAT architectures. Our analysis shows that, unfortunately, no free lunch exists among these three GNN architectures since their performance, as expected, is fully data-dependent.

This motivates the main contribution of the paper, the *learnable graph convolutional attention network* (L-CAT): a novel GNN which, in each layer, automatically interpolates between the three operations by introducing only two scalar parameters. As a result, L-CAT is able to learn the proper operation to apply at each layer, thus combining different layer types in the same GNN architecture while overcoming the need to cross-validate—a process that was prohibitively expensive prior to this work. Our extensive empirical analysis demonstrates the capabilities of L-CAT on a wide range of datasets, outperforming existing baseline GNNs in terms of both performance, and robustness to input noise and network initialization.

## 2 PRELIMINARIES

Assume as input an undirected graph $G = (V, E)$, where $V = [n]$ denotes the set of vertices of the graph, and $E \subseteq V \times V$ the set of edges. Each node $i \in [n]$ is represented by a $d$-dimensional feature vector $\mathbf{X}_i \in \mathbb{R}^d$, and the goal is to produce a set of predictions $\{\hat{\boldsymbol{y}}_i\}_{i=1}^n$. To this end, a message-passing GNN layer yields a representation $\tilde{\boldsymbol{h}}_i \in \mathbb{R}^{d'}$ for each node $i$, by collecting and aggregating the information from each of its neighbors into a single message; and using the aggregated message to update its representation from the previous layer, $\boldsymbol{h}_i \in \mathbb{R}^d$. For the purposes of this work, we can define this operation as the following:

$$\tilde{\boldsymbol{h}}_i = f(\boldsymbol{h}_i') \quad \text{where} \quad \boldsymbol{h}_i' \stackrel{\text{def}}{=} \sum_{j \in N_i^*} \gamma_{ij} \boldsymbol{W}_v \boldsymbol{h}_j \ , \tag{1}$$

where $N_i^*$ is the set of neighbors of node $i$ (including $i$), $\boldsymbol{W}_v \in \mathbb{R}^{d' \times d}$ a learnable matrix, $f$ an elementwise function, and $\gamma_{ij} \in [0, 1]$ are coefficients such that $\sum_j \gamma_{ij} = 1$ for each node $i$.

Let the input features be $\boldsymbol{h}_i^0 = \mathbf{X}_i$, and $\boldsymbol{h}_i^L = \hat{\boldsymbol{y}}_i$ the predictions, then we can readily define a message-passing GNN (Gilmer et al., 2017) as a sequence of $L$ layers as defined above. Depending on the way the coefficients $\gamma_{ij}$ are computed, we identify different GNN flavors.

**Graph convolutional networks (GCNs)** (Kipf & Welling, 2017) are simple yet effective. In short, GCNs compute the average of the messages, i.e., they assign the same coefficient $\gamma_{ij} = 1/|N_i^*|$ to every neighbor:

$$\tilde{\boldsymbol{h}}_i = f(\boldsymbol{h}_i') \quad \text{where} \quad \boldsymbol{h}_i' \stackrel{\text{def}}{=} \frac{1}{|N_i^*|} \sum_{j \in N_i^*} \boldsymbol{W}_v \boldsymbol{h}_j \ , \tag{2}$$

**Graph attention networks** take a different approach. Instead of assigning a fixed value to each coefficient $\gamma_{ij}$, they compute it as a function of the sender and receiver nodes. A general formulation for these models can be written as follows:

$$\tilde{\boldsymbol{h}}_i = f(\boldsymbol{h}_i') \quad \text{where} \quad \boldsymbol{h}_i' \stackrel{\text{def}}{=} \sum_{j \in N_i^*} \gamma_{ij} \boldsymbol{W}_v \boldsymbol{h}_j \quad \text{and} \quad \gamma_{ij} \stackrel{\text{def}}{=} \frac{\exp(\Psi(\boldsymbol{h}_i, \boldsymbol{h}_j))}{\sum_{\ell \in N_i^*} \exp(\Psi(\boldsymbol{h}_i, \boldsymbol{h}_\ell))} \ . \tag{3}$$

Here, $\Psi(\boldsymbol{h}_i, \boldsymbol{h}_j) \stackrel{\text{def}}{=} \alpha(\boldsymbol{W}_q \boldsymbol{h}_i, \boldsymbol{W}_k \boldsymbol{h}_j)$ is known as the *score function* (or *attention architecture*), and provides a score value between the messages $\boldsymbol{h}_i$ and $\boldsymbol{h}_j$ (or more generally, between a learnable mapping of the messages). From these scores, the (attention) coefficients are obtained by normalizing them, such that $\sum_j \gamma_{ij} = 1$. We can find in the literature different attention layers and, throughout this work, we focus on the original GAT (Velickovic et al., 2018) and its extension GATv2 (Brody et al., 2022):

$$\text{GAT:} \quad \Psi(\boldsymbol{h}_i, \boldsymbol{h}_j) = \text{LeakyRelu}\left(\boldsymbol{a}^\top [\boldsymbol{W}_q \boldsymbol{h}_i || \boldsymbol{W}_k \boldsymbol{h}_j]\right) \ , \tag{4}$$

$$\text{GATv2:} \quad \Psi(\boldsymbol{h}_i, \boldsymbol{h}_j) = \boldsymbol{a}^\top \text{LeakyRelu}\left(\boldsymbol{W}_q \boldsymbol{h}_i + \boldsymbol{W}_k \boldsymbol{h}_j\right) \ , \tag{5}$$

where the learnable parameters are now the attention vector $\boldsymbol{a}$; and the matrices $\boldsymbol{W}_q$, $\boldsymbol{W}_k$, and $\boldsymbol{W}_v$. Following previous work (Velickovic et al., 2018; Brody et al., 2022), we assume that these matrices are coupled, i.e., $\boldsymbol{W}_q = \boldsymbol{W}_k = \boldsymbol{W}_v$. Note that the difference between the

two layers lies in the position of the vector $\boldsymbol{a}$: by taking it out of the nonlinearity, Brody et al. (2022) increased the expressiveness of GATv2. Now, the product of $\boldsymbol{a}$ and a weight matrix does not collapse into another vector. More importantly, the addition of two different attention layers will help us show the versatility of the proposed models later in §6.

## 3 Previous work

In recent years, there has been a surge of research in GNNs. Here, we discuss other GNN models, attention mechanisms, and the recent findings on the limitations of GCNs and GATs.

The literature on GNNs is extensive (Wu et al., 2020b; Hamilton et al., 2017a; Battaglia et al., 2018; Lee et al., 2019), and more abstract definitions of a message-passing GNN are possible, leading to other lines of work trying different ways to compute messages, aggregate them, or update the final message (Hamilton et al., 2017b; Xu et al., 2019; Corso et al., 2020). Alternatively, another line of work fully abandons message-passing, working instead with higher-order interactions (Morris et al., 2019). While some of this work is orthogonal—or directly applicable—to the proposed model, in the main paper we focus on convolutional and attention graph layers, as they are the most widely used (and cited) as of today.

While we consider the original GAT (Velickovic et al., 2018) and GATv2 (Brody et al., 2022), our work can be directly applied to any attention model that sticks to the formulation in Eq. 3. For example, some works propose different metrics for the score function, like the dot-product (Brody et al., 2022), cosine similarity (Thekumparampil et al., 2018), or a combination of various functions (Kim & Oh, 2021). Other works introduce transformer-based mechanisms (Vaswani et al., 2017) based on positional encoding (Dwivedi & Bresson, 2020; Kreuzer et al., 2021) or on the set transformer (Wang et al., 2021a). Finally, there also exist attention approaches designed for specific type of graphs, such as relational (Yun et al., 2019; Busbridge et al., 2019) or heterogeneous graphs (Wang et al., 2019b; Hu et al., 2020b).

### 3.1 On the limitations of GCN and GAT networks

Baranwal et al. (2021) studied classification on a simple stochastic block model, showing that, when the graph is neither too sparse nor noisy, applying one layer of graph convolution increases the regime in which the data is linearly separable. However, this result is highly sensitive to the graph structure, as convolutions essentially collapse the data to the same value in the presence of enough noise. More recently, Fountoulakis et al. (2022) showed that GAT is able to remedy the above issue, and provides perfect node separability regardless of the noise level in the graph. However, a classical argument (see Anderson (2003)) states that *in this particular setting* a linear classifier already achieves perfect separability. These works, in summary, showed scenarios for which GCNs can be beneficial in the absence of noise, and that GAT can outperform GCNs in other scenarios, leaving open the question of which architecture is preferable in terms of performance.

## 4 Convolved attention: benefits and hurdles

In this section, we propose to combine attention with convolution operations. To motivate it, we complement the results of Fountoulakis et al. (2022), providing a synthetic dataset for which *any* 1-layer GCN fails, but 1-layer GAT does not. Thus, proving a clear distinction between GAT and GCN layers. Besides, we show that convolution helps GAT as long as the graph noise is reasonable. The proofs for the two statements in this section appear in Appendix A and follow similar arguments as in Fountoulakis et al. (2022).

This dataset is based on the *contextual stochastic block model* (CSBM) (Deshpande et al., 2018). Let $\varepsilon_1, \ldots, \varepsilon_n$ be iid. uniform samples from $\{-1, 0, 1\}$. Let $C_k = \{j \in [n] \mid \varepsilon_j = k\}$ for $k \in \{-1, 0, 1\}$. We set the feature vector $\mathbf{X}_i \sim \mathcal{N}(\varepsilon_i \boldsymbol{\mu}, \mathbf{I} \cdot \sigma^2)$ where $\boldsymbol{\mu} \in \mathbb{R}^d$, $\sigma \in \mathbb{R}$, and $\mathbf{I} \in \{0, 1\}^{d \times d}$ is the identity matrix. For a given pair $p, q \in [0, 1]$ we consider the stochastic adjacency matrix $\mathbf{A} \in \{0, 1\}^{n \times n}$ defined as follows: for $i, j \in [n]$ in the same class (*intra-edge*), we set $a_{ij} \sim \text{Ber}(p)$;[1] for $i, j$ in different classes (*inter-edge*), we set $a_{ij} \sim \text{Ber}(q)$. We denote

---

[1] Ber($\cdot$) denote the Bernoulli distribution.

by $(\mathbf{X}, \mathbf{A}) \sim \mathsf{CSBM}(n, p, q, \boldsymbol{\mu}, \sigma^2)$ a sample obtained according to the above random process. Our task is then to distinguish (or separate) nodes from $C_0$ vs. $C_{-1} \cup C_1$.

Note that, in general, it is impossible to separate $C_0$ from $C_{-1} \cup C_1$ with a linear classifier and, using one convolutional layer is detrimental for node classification on the CSBM:[2] although the convolution brings the means closer and shrinks the variance, the geometric structure of the problem does not change. On the other hand, we prove that GAT is able to achieve perfect node separability when the graph is not too sparse:

**Theorem 1.** Suppose that $p, q = \Omega(\log^2 n/n)$ and $\|\boldsymbol{\mu}\|_2 = \omega(\sigma\sqrt{\log n})$. Then, there exists a choice of attention architecture $\Psi$ such that, with probability at least $1 - o_n(1)$ over the data $(\mathbf{X}, \mathbf{A}) \sim \mathsf{CSBM}(n, p, q, \boldsymbol{\mu}, \sigma^2)$, GAT separates nodes $C_0$ from $C_1 \cup C_{-1}$.

Moreover, we show using methods from Baranwal et al. (2021), that the above classification threshold $\|\boldsymbol{\mu}\|$ can be improved when the graph noise is reasonable. Specifically, *by applying convolution prior to the attention score*, the variance of the data is greatly reduced, and if the graph is not too noisy, the operation dramatically lowers the bound in Thm. 1. We exploit this insight by introducing the *graph convolutional attention layer* (CAT):

$$\Psi(\boldsymbol{h}_i, \boldsymbol{h}_j) = \alpha(\boldsymbol{W}\tilde{\boldsymbol{h}}_i, \boldsymbol{W}\tilde{\boldsymbol{h}}_j) \quad \text{where} \quad \tilde{\boldsymbol{h}}_i = \frac{1}{|N_i^*|} \sum_{\ell \in N_i^*} \boldsymbol{h}_\ell \,, \tag{6}$$

where $\tilde{\boldsymbol{h}}_i$ are the convolved features of the neighborhood of node $i$. As we show now, CAT improves over GAT by combining convolutions with attention, when the graph noise is low.

**Corollary 2.** Suppose $p, q = \Omega(\log^2 n/n)$ and $\|\boldsymbol{\mu}\| \geq \omega\left(\sigma\sqrt{\frac{(p+2q)\log n}{n(p-q)^2}}\right)$. Then, there is a choice of attention architecture $\Psi$ such that CAT separates nodes $C_0$ from $C_1 \cup C_{-1}$, with probability at least $1 - o(1)$ over the data $(\mathbf{X}, \mathbf{A}) \sim \mathsf{CSBM}(n, p, q, \boldsymbol{\mu}, \sigma^2)$.

The above proposition shows that under the $\mathsf{CSBM}$ data model, convolving prior to attention changes the regime for perfect node separability by a factor of $|p - q|\sqrt{n/(p + 2q)}$. This is desirable when $|p - q|\sqrt{n/(p + 2q)} > 1$, since the regime for perfect classification is increased. Otherwise, applying convolution prior to attention reduces the regime for perfect separability. Therefore, it is not always clear whether convolving prior to attention is beneficial.

## 5   L-CAT: LEARNING TO INTERPOLATE

From the previous analysis, we can conclude that it is hard to know *a priori* whether attention, convolution, or convolved attention, will perform the best. In this section, we argue that this issue can be easily overcome by learning to interpolate between the three.

First, note that GCN and GAT only differ in that GCN weighs all neighbors equally (Eq. 2) and, the more similar the attention scores are (Eq. 3), the more uniform the coefficients $\gamma_{ij}$ are. Thus, we can interpolate between GCN and GAT by introducing a learnable parameter. Similarly, the formulation of GAT (Eq. 3) and CAT (Eq. 6) differ in the convolution within the score, which can be interpolated with another learnable parameter.

Following this observation, we propose the *learnable convolutional attention layer* (L-CAT), which can be formulated as an attention layer with the following score:

$$\Psi(\boldsymbol{h}_i, \boldsymbol{h}_j) = \lambda_1 \cdot \alpha(\boldsymbol{W}\tilde{\boldsymbol{h}}_i, \boldsymbol{W}\tilde{\boldsymbol{h}}_j) \quad \text{where} \quad \tilde{\boldsymbol{h}}_i = \frac{\boldsymbol{h}_i + \lambda_2 \sum_{\ell \in N_i} \boldsymbol{h}_\ell}{1 + \lambda_2 |N_i|} \,, \tag{7}$$

where $\lambda_1, \lambda_2 \in [0, 1]$. As mentioned before, this formulation lets L-CAT learn to interpolate between GCN ($\lambda_1 = 0$), GAT ($\lambda_1 = 1$ and $\lambda_2 = 0$), and CAT ($\lambda_1 = 1$ and $\lambda_2 = 1$).

L-CAT enables a number of non-trivial benefits. Not only can it switch between existing layers, but it also learns the amount of attention necessary for each use-case. Moreover, by

---

[2]We note that this problem can be easily solved by two layers of GCN (Baranwal et al., 2022).

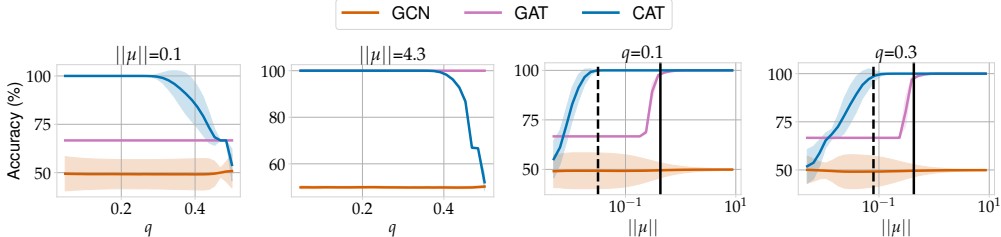

Figure 1: Synthetic data results. The left-most plots show accuracy as we vary the noise level $q$ for $\|\boldsymbol{\mu}\| = 0.1$ and $\|\boldsymbol{\mu}\| = 4.3$. The right-most plots show the accuracy as we change the norm of the means $\|\boldsymbol{\mu}\|$ for $q = 0.1$ and $q = 0.3$. We use two vertical lines to present the classification threshold stated in Thm. 1 (solid line) and Cor. 2 (dashed line).

comprising the three layers in a single learnable formulation, it removes the necessity of cross-validating the type of layer, as their performance is data-dependent (see §§3.1 and 4). Remarkably, it allows to easily combine different layer types within the same architecture.

While we focus on GCN, L-CAT can be easily used with other GNN architectures such as PNA (Corso et al., 2020) and GCNII (Chen et al., 2020), as L-CAT interpolates between two different adjacency matrices. For further details and results, refer to App. F.

## 6 Experiments

In this section, we first validate our theoretical findings on synthetic data (§6.1). Then, we show through various node classification tasks that (L-)CAT is as competitive as the baseline models (§6.2). Lastly, we move to more demanding scenarios from the Open Graph Benchmark (Hu et al., 2020a), demonstrating that L-CAT is a more flexible and robust alternative to its baseline methods (§6.3) that reduces the need of cross-validating without giving up on performance. Refer to Apps. B to F for details and additional results. The code to reproduce the experiments can be found at `https://github.com/psanch21/LCAT`.

### 6.1 Synthetic data

First, we empirically validate our theoretical results (Thm. 1 and Cor. 2). We aim to better understand the behavior of each layer as the properties of the data change, i.e., the noise level $q$ (proportion of inter-edges) and the distance between the means of consecutive classes $\|\boldsymbol{\mu}\|$. We provide extra results and additional experiments in App. B.

**Experimental setup.** As data model, we use the proposed CSBM (see §4) with $n = 10000$, $p = 0.5$, $\sigma = 0.1$, and $d = n/\left(5\log^2(n)\right)$. All results are averaged over 50 runs, and parameters are set as described in App. A. We conduct two experiments to assess the sensitivity to structural noise. First, we vary the noise level $q$ between 0 and 0.5, leaving the mean vector $\boldsymbol{\mu}$ fixed. We test two values of $\|\boldsymbol{\mu}\|$: the first corresponds to the *easy* regime ($\|\boldsymbol{\mu}\| = 10\sigma\sqrt{2\log n}$) where classes are far apart; and the second correspond to the *hard* regime ($\|\boldsymbol{\mu}\| = \sigma$) where clusters are close. In the second experiment we instead sweep $\|\boldsymbol{\mu}\|$ in the range $\left[\sigma/20, 20\sigma\sqrt{2\log n}\right]$, covering the transition from hard (small $\|\boldsymbol{\mu}\|$) to easy (large $\|\boldsymbol{\mu}\|$) settings. Here, we fix $q$ to 0.1 (low noise) and 0.3 (high noise). In both cases, we compare the behavior of 1-layer GAT and CAT, and include GCN as the baseline.

**Results.** The two left-most plots of Fig. 1 show node classification performance for the hard and easy regimes, respectively, as we vary the noise level $q$. In the hard regime, we observe that GAT is unable to achieve separation for any value of $q$, whereas CAT achieves perfect classification when $q$ is small enough. This exemplifies the advantage of CAT over GAT as stated in Cor. 2. When the distance between the means is large enough, we see that GAT achieves perfect results independently of $q$, as stated in Thm. 1. In contrast, when CAT fails to satisfy the condition in Cor. 2 (as we increase $q$), it achieves inferior performance.

The right-most part of Fig. 1 shows the results when we fix $q$ and sweep $\|\boldsymbol{\mu}\|$. In these two plots, we can appreciate the transition in the accuracy of both GAT and CAT as a function of $\|\boldsymbol{\mu}\|$. We observe that GAT achieves perfect accuracy when the distance between the means satisfies the condition in Thm. 1 (solid vertical line in Fig. 1). Moreover, we can see

Table 1: Test accuracy (%) of the considered models for different datasets (sorted by their average node degree), and averaged over ten runs. Bold numbers are statistically different to their baseline model ($\alpha = 0.05$). Best average performance is underlined.

| Dataset | Amazon Computers | Amazon Photo | GitHub | Facebook PagePage | Coauthor Physics | TwitchEN |
|---|---|---|---|---|---|---|
| Avg. Deg. | 35.76 | 31.13 | 15.33 | 15.22 | 14.38 | 10.91 |
| GCN | $\underline{90.59 \pm 0.36}$ | $\underline{95.13 \pm 0.57}$ | $84.13 \pm 0.44$ | $94.76 \pm 0.19$ | $\underline{96.36 \pm 0.10}$ | $57.83 \pm 1.13$ |
| GAT | $89.59 \pm 0.61$ | $94.02 \pm 0.66$ | $83.31 \pm 0.18$ | $94.16 \pm 0.48$ | $96.36 \pm 0.10$ | $57.59 \pm 1.20$ |
| CAT | $\mathbf{90.58 \pm 0.40}$ | $\mathbf{94.77 \pm 0.47}$ | $\mathbf{84.11 \pm 0.66}$ | $\mathbf{94.71 \pm 0.30}$ | $96.40 \pm 0.10$ | $58.09 \pm 1.61$ |
| L-CAT | $\mathbf{90.34 \pm 0.47}$ | $\mathbf{94.93 \pm 0.37}$ | $84.05 \pm 0.70$ | $\mathbf{94.81 \pm 0.25}$ | $96.35 \pm 0.10$ | $57.88 \pm 2.07$ |
| GATv2 | $89.49 \pm 0.53$ | $93.47 \pm 0.62$ | $82.92 \pm 0.45$ | $93.44 \pm 0.30$ | $96.24 \pm 0.19$ | $57.70 \pm 1.17$ |
| CATv2 | $\mathbf{90.44 \pm 0.46}$ | $\mathbf{94.81 \pm 0.55}$ | $\mathbf{84.10 \pm 0.88}$ | $\mathbf{94.27 \pm 0.31}$ | $96.34 \pm 0.12$ | $57.99 \pm 2.02$ |
| L-CATv2 | $\mathbf{90.33 \pm 0.44}$ | $\mathbf{94.79 \pm 0.61}$ | $\underline{\mathbf{84.31 \pm 0.59}}$ | $\mathbf{94.44 \pm 0.39}$ | $96.29 \pm 0.13$ | $57.89 \pm 1.53$ |

the improvement CAT obtains over GAT. Indeed, when $\|\boldsymbol{\mu}\|$ satisfies the conditions of Cor. 2 (dashed vertical line in Fig. 1), the classification threshold is improved. As we increase $q$ we see that the gap between the two vertical lines decreases, which means that the improvement of CAT over GAT decreases as $q$ increments, exactly as stated in Cor. 2.

## 6.2 REAL DATA

We study now the performance of the proposed models in a comprehensive set of real-world experiments, in order to gain further insights of the settings in which they excel. Specifically, we found CAT and L-CAT to outperform their baselines as the average node degree increases. For a detailed description of the datasets and additional results, refer to Apps. C, D and F.

**Models.** We consider as baselines a simple GCN layer (Kipf & Welling, 2017), the original GAT layer (Velickovic et al., 2018) and its recent extension, GATv2 (Brody et al., 2022). Based on the two attention models, we consider their CAT and L-CAT extensions. To ensure fair comparisons, all layers use the same number of parameters and implementation.

**Datasets.** We consider six node classification datasets. The *Facebook/GitHub/TwitchEN* datasets involve social-network graphs (Rozemberczki et al., 2021), whose nodes represent verified pages/developers/streamers; and where the task is to predict the topic/expertise/explicit-language-use of the node. The *Coauthor Physics* dataset (Shchur et al., 2018) represents a co-authorship network whose nodes represent authors, and the task is to infer their main research field. The *Amazon* datasets represent two product-similarity graphs (Shchur et al., 2018), where each node is a product, and the task is to infer its category.

**Experimental setup.** To ensure the best results, we cross-validate all optimization-related hyperparameters for each model using GraphGym (You et al., 2020). All models use four GNN layers with hidden size of 32, and thus have an equal number of parameters. For evaluation, we take the best-validation configuration during training, and report test-set performance. For further details, refer to App. D.

**Results** are presented in Table 1. In contrast with §6.1, we here find GCN to be a strong contender, reinforcing its viability in real-world data despite its simplicity. We observe both CAT and L-CAT not only holding up the performance with respect to their baselines models for all datasets, but in most cases also improving the test accuracy in a statistically significant manner. These results validate the effectiveness of CAT as a GNN layer, and show the viability of *L-CAT as a drop-in replacement*, achieving good results on all datasets.

As explained in §4, CAT differs from a usual GAT in that the score is computed with respect to the convolved features. Intuitively, this means that CAT should excel in those settings where nodes are better connected, allowing CAT to extract more information from their neighborhoods. Indeed, in the inset figure we can observe the improvement in accuracy of CAT with respect to its baseline model, as a function of the average node degree of the dataset, and the linear regression fit of these results (dashed line). This plot includes all datasets

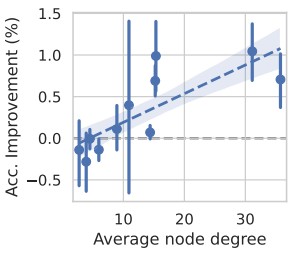

Table 2: Test performance of the considered models on four OGB datasets, averaged over five runs. Bold numbers are statistically different to their baseline model ($\alpha = 0.05$). Best average performance is underlined. Left table: accuracy (%); right table: AUC-ROC (%).

| Dataset | *arxiv* | *products* | *mag* | *proteins* |
|---------|---------|------------|-------|------------|
| GCN | $71.58 \pm 0.20$ | $74.12 \pm 1.20$ | $\underline{32.77 \pm 0.36}$ | $\underline{80.10 \pm 0.55}$ |
| GAT | $71.58 \pm 0.16$ | $78.53 \pm 0.91$ | $32.15 \pm 0.31$ | $79.08 \pm 1.47$ |
| CAT | $\mathbf{72.14 \pm 0.21}$ | $\mathbf{77.38 \pm 0.36}$ | $31.98 \pm 0.46$ | $73.26 \pm 1.65$ |
| L-CAT | $\underline{71.99 \pm 0.08}$ | $77.19 \pm 1.11$ | $32.47 \pm 0.38$ | $79.63 \pm 0.71$ |
| GATv2 | $71.73 \pm 0.24$ | $76.40 \pm 0.71$ | $32.76 \pm 0.18$ | $78.65 \pm 1.44$ |
| CATv2 | $\mathbf{72.03 \pm 0.09}$ | $74.81 \pm 1.12$ | $\mathbf{32.43 \pm 0.22}$ | $74.33 \pm 0.94$ |
| L-CATv2 | $71.97 \pm 0.22$ | $\mathbf{76.37 \pm 0.92}$ | $32.68 \pm 0.50$ | $\mathbf{79.07 \pm 0.98}$ |

(from the manuscript and App. D), and shows a positive trend between node connectivity and improved performance achieved by CAT.

## 6.3 OPEN GRAPH BENCHMARK

In this section, we assess the robustness of the proposed models, in order to fully understand their benefits. For further details and additional results, refer to App. E.

**Datasets.** We consider four datasets from the OGB suite (Hu et al., 2020a): *proteins*, *products*, *arxiv*, and *mag*. Note that these datasets are significantly larger than those from §6.2 and correspond to more difficult tasks, e.g., *arxiv* is a 40-class classification problem (see Table 5 in App. C for details). This makes them more suitable for the proposed analysis.

**Experimental setup.** We adopt the same experimental setup as Brody et al. (2022) for the *proteins*, *products*, and *mag* datasets. For the *arxiv* dataset, we use instead the example code from OGB (Hu et al., 2020a), as it yields better performance than that of Brody et al. (2022). Just as in §6.2, we compare with GCN (Kipf & Welling, 2017), GAT (Velickovic et al., 2018), GATv2 (Brody et al., 2022), and their CAT and L-CAT counterparts. We cross-validate the number of heads (1 and 8), and select the best-validation models during training. All models are identical except for their $\lambda$ values.

**Results** are summarized in Table 2. Here we do not observe a clear preferred baseline: GCN performs really well in *proteins* and *mag*; GAT excels in *products*; and GATv2 does well in *arxiv* and *mag*. While CAT obtains the best results on *arxiv*, its performance on *proteins* and *products* is significantly worse than the baseline model. Presumably, an excessive amount of inter-edges could explain why convolving the features prior to computing the score is harmful, as seen in §6.1. As we explore in §6.3.2, however, CAT improves over its baseline for most *proteins* scenarios, specially with a single head. In stark contrast, L-CAT performs remarkably well, improving the baseline models in all datasets but *products*—even on those in which CAT fails—demonstrating the adaptability of L-CAT to different scenarios.

To better understand the training dynamics of the models, we plot in Fig. 2a the test accuracy of GCN and the GATv2 models during training on the *arxiv* dataset. Interestingly, despite all models obtaining similar final results, *CATv2 and L-CATv2 drastically improved their convergence speed and stability with respect to GATv2*, matching that of GCN. To understand the behavior of L-CATv2, Fig. 2b shows the evolution of the $\lambda$ parameters. We observe that, to achieve these results, L-CATv2 converged to a GNN network that combines three types of layers: the first layer is a CATv2 layer, taking advantage of the neighboring information; the second layer is a quasi-GCN layer, in which scores are almost uniform and some neighboring information is still used in the score computation; and the third layer is a pure GCN layer, in which all scores are uniformly distributed. It is important to remark that these dynamics are fairly consistent, as L-CATv2 reached the same $\lambda$ values over all five runs.

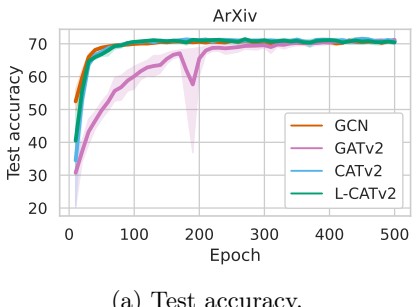
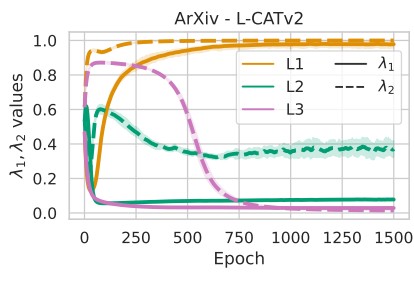

(a) Test accuracy.

(b) Evolution of $\lambda_1, \lambda_2$.

Figure 2: Behavior of GCN and GATv2-based models during training on the *arxiv* dataset. *(a)* CAT and L-CAT converge quicker and more stably than their baseline model. *(b)* L-CAT consistently converges to the same architecture: a CAT →quasi-GCN→GCN network.

### 6.3.1 ROBUSTNESS TO NOISE

One intrinsic aspect of real world data is the existence of noise. In this section, we explore the robustness of the proposed models to different levels of noise, i.e., we attempt to simulate scenarios where there exist measurement inaccuracies in the input features and edges.

**Experimental setup.** We consider the *arxiv* dataset, and the same experimental setup as in §6.3. We conduct two experiments. First, we introduce to the node features additive noise of the form $\mathcal{N}(\mathbf{0}, \mathbf{1}\sigma)$, and consider different levels of noise, $\sigma \in \{0, 0.25, 0.5, 0.75, 1\}$. Then, as in (Brody et al., 2022), we simulate edge noise by adding fake edges with probability $Bern(p)$ for $p \in \{0, 0.1, 0.2, 0.3, 0.4, 0.5\}$.

**Results** are shown in the inset figures for feature (top) and edge noise (bottom), summarizing the performance of all models over five runs and two numbers of heads (1 and 8). Baseline attention models are quite sensitive to feature noise, but are more robust to edge noise, as they can drop inter-class edges (see §3.1). GCNs, as expected, are instead more robust to feature noise, but suffer more in the presence of edge noise. In concordance with the synthetic experiments (see §§4 and 6.1), CAT is able to leverage convolutions as a variance-reduction technique, reducing the variance and improving its robustness to feature noise. Remarkably, L-CAT proves to be the most robust for both types of noise: by adapting the amount of attention used in each layer, it outperforms existing methods and reduces the variance.

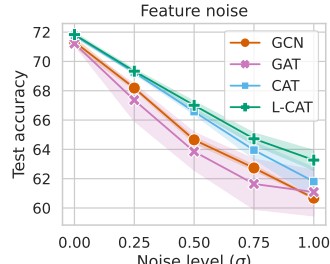

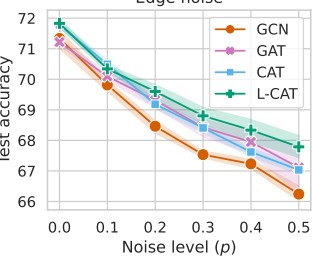

### 6.3.2 ROBUSTNESS TO NETWORK INITIALIZATION

Another important aspect for real-world applications is robustness to network initialization, i.e., the ability to obtain satisfying performance independently of the initial parameters. Otherwise, a practitioner can waste lots of resources trying initilizations or, even worse, give up on a model just because they did not try the initial parameters that yield great results.

**Experimental setup.** We follow once again the same setup for *proteins* as in §6.3. We consider two different network initializations. The first one, *uniform*, uses uniform Glorot initilization (Glorot & Bengio, 2010) with a gain of 1, which is the standard initialization used throughout this work. The second one, *normal*, uses instead normal Glorot initialization (Glorot & Bengio, 2010) with a gain of $\sqrt{2}$. This is the initialization employed on the original GATv2 paper (Brody et al., 2022) exclusively for the *proteins* dataset.

**Results**—segregated by number of heads—are shown in Table 3, while the results for GCN appear in the inset table. These results show that the baseline models perform poorly on the

Table 3: Test AUC-ROC (%) on the *proteins* dataset for attention models with two different network initializations (see §6.3.2), using 1 head (top) and 8 heads (bottom).

|  |  | GAT | CAT | L-CAT | GATv2 | CATv2 | L-CATv2 |
|---|---|---|---|---|---|---|---|
| 1h | *uniform* | $59.73 \pm 3.61$ | $\mathbf{64.32 \pm 2.33}$ | $\mathbf{77.77 \pm 1.28}$ | $59.85 \pm 2.73$ | $\mathbf{64.32 \pm 2.33}$ | $\mathbf{79.08 \pm 0.95}$ |
|  | *normal* | $66.38 \pm 6.94$ | $73.26 \pm 1.65$ | $\mathbf{78.06 \pm 1.25}$ | $69.13 \pm 8.48$ | $74.33 \pm 0.94$ | $\underline{\mathbf{79.07 \pm 0.98}}$ |
| 8h | *uniform* | $72.23 \pm 2.86$ | $73.60 \pm 1.14$ | $\mathbf{78.85 \pm 1.57}$ | $75.21 \pm 1.61$ | $74.16 \pm 1.30$ | $\mathbf{78.77 \pm 0.97}$ |
|  | *normal* | $79.08 \pm 1.47$ | $\mathbf{74.67 \pm 1.15}$ | $\underline{79.63 \pm 0.71}$ | $78.65 \pm 1.44$ | $\mathbf{73.40 \pm 0.56}$ | $79.30 \pm 0.49$ |
|  | average | $69.36 \pm 8.52$ | $73.93 \pm 1.35$ | $78.58 \pm 1.48$ | $70.71 \pm 8.70$ | $71.55 \pm 4.54$ | $\underline{79.05 \pm 0.91}$ |

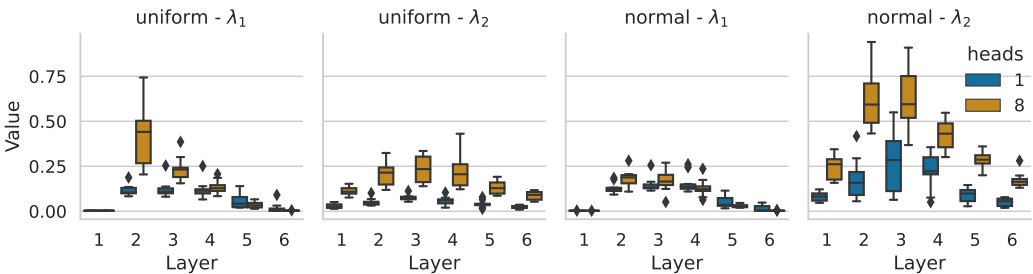

Figure 3: Distribution of $\lambda_1, \lambda_2$ on *proteins* dataset for L-CAT across initializations.

*uniform* initialization. However, this is somewhat alleviated when using 8 heads in the attention models. Moreover, all baselines significantly improve with *normal* initialization, being GCN the best model, and attention models obtaining 79 % accuracy on average with 8 heads.

Compared to the baselines, CAT does a good job and improves the performance in all cases except for *normal* with 8 heads. Remarkably, L-CAT consistently obtains high accuracy in all scenarios and runs. To emphasize consistency, bottom row shows the average accuracy across runs, showing that L-CAT is clearly more robust to parameter initialization than competing models.

|  | GCN |
|---|---|
| *uniform* | $61.08 \pm 2.56$ |
| *normal* | $80.10 \pm 0.55$ |
| average | $70.59 \pm 10.21$ |

To understand this performance, we inspect the distribution of $\lambda_1, \lambda_2$ for L-CAT in Fig. 3. Here, we can spot a few interesting patterns. Consistently, the first and last layers are always GCNs, while the inner layers progressively admit less attention. Second, the number of heads affects the amount of attention allowed in the network; the more heads, the more expressive the layer tends to be, and more attention is permitted. Third, L-CAT adapts to the initialization used: in *uniform*, it allows more attention in the second layer; in *normal*, it allows more attention in the score inputs. These results consolidate the flexibility of L-CAT.

## 7 Conclusions and future work

In this work, we studied how to combine the strengths of convolution and attention layers in GNNs. We proposed CAT, which computes attention with respect to the convolved features, and analyzed its benefits and limitations on a new synthetic dataset. This analysis revealed different regimes where one model is preferred over the others, reinforcing the idea that selecting between GCNs, GATs, and now CATs, is a difficult task. For this reason, we proposed L-CAT, a model which interpolates between the three via two learnable parameters. Extensive experimental results demonstrated the effectiveness of L-CAT, yielding great results while being more robust than other methods. As a result, L-CAT proved to be a viable drop-in replacement that removes the need to cross-validate the layer type.

We strongly believe learnable interpolation can get us a long way, and we hope L-CAT to motivate new and exciting work. Specially, we are eager to see L-CAT in real applications, and thus finding out what combining different GNN layers across a model (without the hurdle of cross-validating all layer combinations) can lead to in the real-world.

ETHIC STATEMENT

Given the nature of this work, we do not see any direct ethical concerns. On the contrary, L-CAT eases the application of GNNs to the practitioner, and removes the need of cross-validating the layer type, which can potentially benefit other areas and applications, as GNNs have already proven.

REPRODUCIBILITY STATEMENT

For the theoretical results, we describe the data model used in §4, and provide all the detailed proofs in App. A. For the experimental results, we include in the supplementary material the necessary code and scripts required to reproduce our experiments, and all required datasets are freely available. Complete details about the experimental setup can be found in Apps. B, D and E, and we report in our results the mean and standard deviation computed using five trials or more. In addition, we highlight in bold the results that are statistically significant. Moreover, we include details of computational resources used for the all sets of experiments in App. B, App. D and App. E.

ACKNOWLEDGEMENTS

We would like to thank Batuhan Koyuncu, Jonas Klesen, Miriam Rateike, and Maryam Meghdadi for helpful feedback and discussions. Pablo Sánchez Martín thanks the German Research Foundation through the Cluster of Excellence "Machine Learning – New Perspectives for Science", EXC 2064/1, project number 390727645 for generous funding support. The authors thank the International Max Planck Research School for Intelligent Systems (IMPRS-IS) for supporting Pablo Sánchez Martín.

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

# Appendix

## Table of Contents

## A  Theoretical results

### A.1  A hard example for GCN

In this subsection, we present a dataset and classification task for which GCN performs poorly. Note that we follow the similar techniques and notation as (Fountoulakis et al., 2022), as described in the main paper.

We recall our data model. Fix $n, d \in \mathbb{N}$ and let $\varepsilon_1, \ldots, \varepsilon_n$ be i.i.d uniformly sampled from $\{-1, 0, 1\}$. Let $C_k = \{j \in [n] \mid \varepsilon_j = k\}$ for $k \in \{-1, 0, 1\}$. For each index $i \in [n]$, we set the feature vector $\mathbf{X}_i \in \mathbb{R}^d$ as $\mathbf{X}_i \sim \mathcal{N}(\varepsilon_i \cdot \boldsymbol{\mu}, \mathbf{I} \cdot \sigma^2)$, where $\boldsymbol{\mu} \in \mathbb{R}^d$, $\sigma \in \mathbb{R}$ and $\mathbf{I} \in \{0, 1\}^{d \times d}$ is the identity matrix. For a given pair $p, q \in [0, 1]$ we consider the stochastic adjacency matrix $\mathbf{A} \in \{0, 1\}^{n \times n}$ defined as follows. For $i, j \in [n]$ in the same class, we set $a_{ij} \sim \mathrm{Ber}(p)$, and if $i, j$ are in different classes, we set $a_{ij} \sim \mathrm{Ber}(q)$. We let $\mathbf{D} \in \mathbb{R}^{n \times n}$ be a diagonal matrix containing the degrees of the vertices. We denote by $(\mathbf{X}, \mathbf{A}) \sim \mathsf{CSBM}(n, p, q, \boldsymbol{\mu}, \sigma^2)$ a sample obtained according to the above random process.

The task we wish to solve is classifying $C_0$ vs $C_{-1} \cup C_1$. Namely, we want our model $\varphi$ to satisfy $\varphi(\mathbf{X}_i) < 0$ if and only if $i \in C_0$. Moreover, note that the posed problem *is not linearly classifiable*.

To this end, we start by stating an assumption on the choice of parameters. This assumption is necessary to achieve degree concentration in the graph.

**Assumption 1.** $p, q = \Omega(\log^2 n/n)$ .

We now show the distribution of the convolved features. The following lemma can be easily obtained using the techniques in Baranwal et al. (2021).

**Lemma 3.** Fix $p, q$ satisfying Assumption 1. With probability at least $1 - o(1)$ over $\mathbf{A}$ and $\{\varepsilon_i\}_i$,
$$(\mathbf{D}^{-1}\mathbf{A}\mathbf{X})_i \sim \mathcal{N}\left(\varepsilon_i \cdot \frac{p-q}{p+2q}\boldsymbol{\mu}, \frac{\sigma^2}{n(p+2q)}\right), \qquad \forall i \in [n].$$

To prove the above lemma, we need the following definition of our high probability event.

**Definition 1.** We define the even $\mathcal{E}$ as the intersection of the following events over $\mathbf{A}$ and $\{\varepsilon_i\}_i$:

1.  $\mathcal{E}_1$ is the event that $|C_0| = \frac{n}{3} \pm O(\sqrt{n \log n})$, $|C_1| = \frac{n}{3} \pm O(\sqrt{n \log n})$ and $|C_{-1}| = \frac{n}{3} \pm O(\sqrt{n \log n})$.
2.  $\mathcal{E}_2$ is the event that for each $i \in [n]$, $\mathbf{D}_{ii} = \frac{n(p+2q)}{3}\left(1 \pm \frac{10}{\sqrt{\log n}}\right)$.
3.  $\mathcal{E}_3$ is the event that for each $i \in [n]$ and $k \in \{-1, 0, 1\}$,
$$|N_i \cap C_k| = \begin{cases} \mathbf{D}_{ii} \cdot \frac{p}{p+2q} \cdot \left(1 \pm \frac{10}{\sqrt{\log n}}\right) & \text{if } i \in C_k \\ \mathbf{D}_{ii} \cdot \frac{q}{p+2q} \cdot \left(1 \pm \frac{10}{\sqrt{\log n}}\right) & \text{if } i \notin C_k \end{cases}.$$

The following lemma is a direct application of Chernoff bound and a union bound.

**Lemma 4.** With probability at least $1 - 1/\mathrm{poly}(n)$ the event $\mathcal{E}$ holds.

**Proof of Lemma 3.** By applying Lemma 4, and conditioned on $\mathcal{E}$, for any $i \in [n]$
$$(\mathbf{D}^{-1}\mathbf{A}\mathbf{X})_i = \frac{1}{\mathbf{D}_{ii}}\sum_{j \in N_i}\mathbf{X}_j = \frac{1}{\mathbf{D}_{ii}}\left(\sum_{j \in N_i \cap C_{-1}}\mathbf{X}_j + \sum_{j \in N_i \cap C_0}\mathbf{X}_j + \sum_{j \in N_i \cap C_1}\mathbf{X}_j\right).$$
Using the definition of $\mathcal{E}$ and properties of Gaussian distributions the lemma follows. $\square$

Lemma 3 shows that essentially, the convolution reduced the variance and moved the means closer, but the structure of the problem stayed exactly the same. Therefore, one layer of GCN cannot separate $C_0$ from $C_{-1} \cup C_1$ with high probability.

## A.2 A SOLUTION FOR GAT AND CAT

In what follows, we show that GAT is able to handle the above classification task easily when the distance between the means is large enough. Then, we show how the additional convolution on the inputs to the score function improves the regime of perfect classification when the graph is not too noisy. Our main technical lemma considers a specific attention architecture and characterize the attention scores for our data model.

**Lemma 5.** Suppose that $p, q$ satisfy Assumption 1, $\|\boldsymbol{\mu}\| \geq \omega\left(\sigma\sqrt{\log n}\right)$, fix the LeakyRelu constant $\beta \in (0, 1)$ and $R \in \mathbb{R}$. Then, there exists a choice of attention architecture $\Psi$ such that with probability at least $1 - o_n(1)$ over the data $(\mathbf{X}, \mathbf{A}) \sim \mathsf{CSBM}(n, p, q, \boldsymbol{\mu}, \sigma^2)$ the following holds.

$$\Psi(\mathbf{X}_i, \mathbf{X}_j) = \begin{cases} 10R\beta\|\boldsymbol{\mu}\|(1 \pm o(1)) & \text{if } i, j \in C_1^2 \\ -2R\|\boldsymbol{\mu}\|(1 + 2\beta)(1 \pm o(1)) & \text{if } i, j \in C_{-1}^2 \\ -2R\|\boldsymbol{\mu}\|(1 + 5\beta)(1 \pm o(1)) & \text{if } i \in C_1, \, j \in C_{-1} \\ 10R\beta\|\boldsymbol{\mu}\|(1 \pm o(1)) & \text{if } i \in C_{-1}, \, j \in C_1 \\ -\frac{R}{2}\|\boldsymbol{\mu}\|(1 - 21\beta)(1 \pm o(1)) & \text{if } i \in C_0, \, j \in C_1 \\ -\frac{R}{2}\|\boldsymbol{\mu}\|(1 - 11\beta)(1 \pm o(1)) & \text{if } i \in C_0, \, j \in C_{-1} \\ -\frac{R}{2}\|\boldsymbol{\mu}\|(1 - 5\beta)(1 \pm o(1)) & \text{if } i \in C_1, \, j \in C_0 \\ -\frac{R}{2}\|\boldsymbol{\mu}\|(1 - 5\beta)(1 \pm o(1)) & \text{if } i \in C_{-1}, \, j \in C_0 \\ 2R\beta\|\boldsymbol{\mu}\|(1 \pm o(1)) & \text{if } i, j \in C_0^2 \end{cases}.$$

**Proof.** We consider as an ansatz the following two layer architecture $\Psi$.

$$\tilde{\boldsymbol{w}} \stackrel{\text{def}}{=} \frac{\boldsymbol{\mu}}{\|\boldsymbol{\mu}\|}, \qquad \mathbf{S} \stackrel{\text{def}}{=} \begin{bmatrix} 1 & 1 \\ -1 & -1 \\ 1 & -1 \\ -1 & 1 \\ 0 & 1 \\ 1 & 0 \\ 0 & -1 \\ -1 & 0 \end{bmatrix}, \qquad \boldsymbol{b} \stackrel{\text{def}}{=} \begin{bmatrix} -3/2 \\ -3/2 \\ -3/2 \\ -3/2 \\ -1/2 \\ -1/2 \\ -1/2 \\ -1/2 \end{bmatrix} \cdot \|\boldsymbol{\mu}\|, \qquad \boldsymbol{r} \stackrel{\text{def}}{=} R \cdot \begin{bmatrix} 2 \\ -2 \\ -2 \\ 2 \\ -1 \\ -1 \\ -1 \\ -1 \end{bmatrix},$$

where $R > 0$ is an arbitrary scaling parameter. The output of the attention model is defined as

$$\Psi(\mathbf{X}_i, \mathbf{X}_j) \stackrel{\text{def}}{=} \boldsymbol{r}^T \cdot \text{LeakyRelu}\left(\mathbf{S} \cdot \begin{bmatrix} \tilde{\boldsymbol{w}}^T \mathbf{X}_i \\ \tilde{\boldsymbol{w}}^T \mathbf{X}_j \end{bmatrix} + \boldsymbol{b}\right).$$

Let $\boldsymbol{\Delta}_{ij} \stackrel{\text{def}}{=} \mathbf{S} \cdot \begin{bmatrix} \tilde{\boldsymbol{w}}^T \mathbf{X}_i \\ \tilde{\boldsymbol{w}}^T \mathbf{X}_j \end{bmatrix} + \boldsymbol{b} \in \mathbb{R}^8$, and note that for each element $t \in [8]$ of $\boldsymbol{\Delta}_{ij}$, we have that $(\boldsymbol{\Delta}_{ij})_t = \mathbf{S}_{t,1}\tilde{\boldsymbol{w}}^T \mathbf{X}_i + \mathbf{S}_{t,2}\tilde{\boldsymbol{w}}^T \mathbf{X}_j + \boldsymbol{b}_t$. Note that the random variable $(\boldsymbol{\Delta}_{ij})_t$ is distributed as follows:

$$(\boldsymbol{\Delta}_{ij})_t \sim \begin{cases} \mathcal{N}\left((\mathbf{S}_{t,1} + \mathbf{S}_{t,2})\tilde{\boldsymbol{w}}^T \boldsymbol{\mu} + \boldsymbol{b}_t, \, \|\mathbf{S}_{t,*}\|^2\sigma^2\right) & \text{if } i, j \in C_1^2 \\ \mathcal{N}\left(-(\mathbf{S}_{t,1} + \mathbf{S}_{t,2})\tilde{\boldsymbol{w}}^T \boldsymbol{\mu} + \boldsymbol{b}_t, \, \|\mathbf{S}_{t,*}\|^2\sigma^2\right) & \text{if } i, j \in C_{-1}^2 \\ \mathcal{N}\left((\mathbf{S}_{t,1} - \mathbf{S}_{t,2})\tilde{\boldsymbol{w}}^T \boldsymbol{\mu} + \boldsymbol{b}_t, \, \|\mathbf{S}_{t,*}\|^2\sigma^2\right) & \text{if } i \in C_1, \, j \in C_{-1} \\ \mathcal{N}\left(-(\mathbf{S}_{t,1} - \mathbf{S}_{t,2})\tilde{\boldsymbol{w}}^T \boldsymbol{\mu} + \boldsymbol{b}_t, \, \|\mathbf{S}_{t,*}\|^2\sigma^2\right) & \text{if } i \in C_{-1}, \, j \in C_1 \\ \mathcal{N}\left(\mathbf{S}_{t,2}\tilde{\boldsymbol{w}}^T \boldsymbol{\mu} + \boldsymbol{b}_t, \, \|\mathbf{S}_{t,*}\|^2\sigma^2\right) & \text{if } i \in C_0, \, j \in C_1 \\ \mathcal{N}\left(-\mathbf{S}_{t,2}\tilde{\boldsymbol{w}}^T \boldsymbol{\mu} + \boldsymbol{b}_t, \, \|\mathbf{S}_{t,*}\|^2\sigma^2\right) & \text{if } i \in C_0, \, j \in C_{-1} \\ \mathcal{N}\left(\mathbf{S}_{t,1}\tilde{\boldsymbol{w}}^T \boldsymbol{\mu} + \boldsymbol{b}_t, \, \|\mathbf{S}_{t,*}\|^2\sigma^2\right) & \text{if } i \in C_1, \, j \in C_0 \\ \mathcal{N}\left(-\mathbf{S}_{t,1}\tilde{\boldsymbol{w}}^T \boldsymbol{\mu} + \boldsymbol{b}_t, \, \|\mathbf{S}_{t,*}\|^2\sigma^2\right) & \text{if } i \in C_{-1}, \, j \in C_0 \\ \mathcal{N}\left(\boldsymbol{b}_t, \, \|\mathbf{S}_{t,*}\|^2\sigma^2\right) & \text{if } i, j \in C_0^2 \end{cases}.$$

Therefore, for a fixed $i, j \in [n]^2$ we have that the entries of $\mathbf{\Delta}_{ij}$ are distributed as follows (where we use $\mathcal{N}^y_x$ as abbreviation for the Gaussian $\mathcal{N}(x, y)$)

$$\left[ \mathcal{N}^{4\sigma^2}_{\frac{\|\boldsymbol{\mu}\|}{2}} \quad \mathcal{N}^{4\sigma^2}_{\frac{-7\|\boldsymbol{\mu}\|}{2}} \quad \mathcal{N}^{4\sigma^2}_{-\frac{3\|\boldsymbol{\mu}\|}{2}} \quad \mathcal{N}^{4\sigma^2}_{-\frac{3\|\boldsymbol{\mu}\|}{2}} \quad \mathcal{N}^{\sigma^2}_{\frac{\|\boldsymbol{\mu}\|}{2}} \quad \mathcal{N}^{\sigma^2}_{\frac{\|\boldsymbol{\mu}\|}{2}} \quad \mathcal{N}^{\sigma^2}_{-\frac{3\|\boldsymbol{\mu}\|}{2}} \quad \mathcal{N}^{\sigma^2}_{-\frac{3\|\boldsymbol{\mu}\|}{2}} \right] \quad \text{for } i, j \in C_1^2,$$

$$\left[ \mathcal{N}^{4\sigma^2}_{-\frac{7\|\boldsymbol{\mu}\|}{2}} \quad \mathcal{N}^{4\sigma^2}_{\frac{\|\boldsymbol{\mu}\|}{2}} \quad \mathcal{N}^{4\sigma^2}_{-\frac{3\|\boldsymbol{\mu}\|}{2}} \quad \mathcal{N}^{4\sigma^2}_{-\frac{3\|\boldsymbol{\mu}\|}{2}} \quad \mathcal{N}^{\sigma^2}_{-\frac{3\|\boldsymbol{\mu}\|}{2}} \quad \mathcal{N}^{\sigma^2}_{-\frac{3\|\boldsymbol{\mu}\|}{2}} \quad \mathcal{N}^{\sigma^2}_{\frac{\|\boldsymbol{\mu}\|}{2}} \quad \mathcal{N}^{\sigma^2}_{\frac{\|\boldsymbol{\mu}\|}{2}} \right] \quad \text{for } i, j \in C_{-1}^2,$$

$$\left[ \mathcal{N}^{4\sigma^2}_{-\frac{3\|\boldsymbol{\mu}\|}{2}} \quad \mathcal{N}^{4\sigma^2}_{-\frac{3\|\boldsymbol{\mu}\|}{2}} \quad \mathcal{N}^{4\sigma^2}_{\frac{\|\boldsymbol{\mu}\|}{2}} \quad \mathcal{N}^{4\sigma^2}_{-\frac{7\|\boldsymbol{\mu}\|}{2}} \quad \mathcal{N}^{\sigma^2}_{-\frac{3\|\boldsymbol{\mu}\|}{2}} \quad \mathcal{N}^{\sigma^2}_{\frac{\|\boldsymbol{\mu}\|}{2}} \quad \mathcal{N}^{\sigma^2}_{\frac{\|\boldsymbol{\mu}\|}{2}} \quad \mathcal{N}^{\sigma^2}_{-\frac{3\|\boldsymbol{\mu}\|}{2}} \right] \quad \text{for } i, j \in C_1 \times C_{-1},$$

$$\left[ \mathcal{N}^{4\sigma^2}_{-\frac{3\|\boldsymbol{\mu}\|}{2}} \quad \mathcal{N}^{4\sigma^2}_{-\frac{3\|\boldsymbol{\mu}\|}{2}} \quad \mathcal{N}^{4\sigma^2}_{-\frac{7\|\boldsymbol{\mu}\|}{2}} \quad \mathcal{N}^{4\sigma^2}_{\frac{\|\boldsymbol{\mu}\|}{2}} \quad \mathcal{N}^{\sigma^2}_{\frac{\|\boldsymbol{\mu}\|}{2}} \quad \mathcal{N}^{\sigma^2}_{-\frac{3\|\boldsymbol{\mu}\|}{2}} \quad \mathcal{N}^{\sigma^2}_{-\frac{3\|\boldsymbol{\mu}\|}{2}} \quad \mathcal{N}^{\sigma^2}_{\frac{\|\boldsymbol{\mu}\|}{2}} \right] \quad \text{for } i, j \in C_{-1} \times C_1,$$

$$\left[ \mathcal{N}^{4\sigma^2}_{-\frac{\|\boldsymbol{\mu}\|}{2}} \quad \mathcal{N}^{4\sigma^2}_{-\frac{5\|\boldsymbol{\mu}\|}{2}} \quad \mathcal{N}^{4\sigma^2}_{-\frac{5\|\boldsymbol{\mu}\|}{2}} \quad \mathcal{N}^{4\sigma^2}_{-\frac{\|\boldsymbol{\mu}\|}{2}} \quad \mathcal{N}^{\sigma^2}_{\frac{\|\boldsymbol{\mu}\|}{2}} \quad \mathcal{N}^{\sigma^2}_{-\frac{\|\boldsymbol{\mu}\|}{2}} \quad \mathcal{N}^{\sigma^2}_{-\frac{3\|\boldsymbol{\mu}\|}{2}} \quad \mathcal{N}^{\sigma^2}_{-\frac{\|\boldsymbol{\mu}\|}{2}} \right] \quad \text{for } i, j \in C_0 \times C_1,$$

$$\left[ \mathcal{N}^{4\sigma^2}_{-\frac{5\|\boldsymbol{\mu}\|}{2}} \quad \mathcal{N}^{4\sigma^2}_{-\frac{\|\boldsymbol{\mu}\|}{2}} \quad \mathcal{N}^{4\sigma^2}_{-\frac{\|\boldsymbol{\mu}\|}{2}} \quad \mathcal{N}^{4\sigma^2}_{-\frac{5\|\boldsymbol{\mu}\|}{2}} \quad \mathcal{N}^{\sigma^2}_{-\frac{3\|\boldsymbol{\mu}\|}{2}} \quad \mathcal{N}^{\sigma^2}_{-\frac{\|\boldsymbol{\mu}\|}{2}} \quad \mathcal{N}^{\sigma^2}_{\frac{\|\boldsymbol{\mu}\|}{2}} \quad \mathcal{N}^{\sigma^2}_{-\frac{\|\boldsymbol{\mu}\|}{2}} \right] \quad \text{for } i, j \in C_0 \times C_{-1},$$

$$\left[ \mathcal{N}^{4\sigma^2}_{-\frac{\|\boldsymbol{\mu}\|}{2}} \quad \mathcal{N}^{4\sigma^2}_{-\frac{5\|\boldsymbol{\mu}\|}{2}} \quad \mathcal{N}^{4\sigma^2}_{-\frac{\|\boldsymbol{\mu}\|}{2}} \quad \mathcal{N}^{4\sigma^2}_{-\frac{5\|\boldsymbol{\mu}\|}{2}} \quad \mathcal{N}^{\sigma^2}_{-\frac{\|\boldsymbol{\mu}\|}{2}} \quad \mathcal{N}^{\sigma^2}_{\frac{\|\boldsymbol{\mu}\|}{2}} \quad \mathcal{N}^{\sigma^2}_{-\frac{\|\boldsymbol{\mu}\|}{2}} \quad \mathcal{N}^{\sigma^2}_{-\frac{3\|\boldsymbol{\mu}\|}{2}} \right] \quad \text{for } i, j \in C_1 \times C_0,$$

$$\left[ \mathcal{N}^{4\sigma^2}_{-\frac{5\|\boldsymbol{\mu}\|}{2}} \quad \mathcal{N}^{4\sigma^2}_{-\frac{\|\boldsymbol{\mu}\|}{2}} \quad \mathcal{N}^{4\sigma^2}_{-\frac{5\|\boldsymbol{\mu}\|}{2}} \quad \mathcal{N}^{4\sigma^2}_{-\frac{\|\boldsymbol{\mu}\|}{2}} \quad \mathcal{N}^{\sigma^2}_{-\frac{\|\boldsymbol{\mu}\|}{2}} \quad \mathcal{N}^{\sigma^2}_{-\frac{3\|\boldsymbol{\mu}\|}{2}} \quad \mathcal{N}^{\sigma^2}_{-\frac{\|\boldsymbol{\mu}\|}{2}} \quad \mathcal{N}^{\sigma^2}_{\frac{\|\boldsymbol{\mu}\|}{2}} \right] \quad \text{for } i, j \in C_{-1} \times C_0,$$

$$\left[ \mathcal{N}^{4\sigma^2}_{-\frac{3\|\boldsymbol{\mu}\|}{2}} \quad \mathcal{N}^{4\sigma^2}_{-\frac{3\|\boldsymbol{\mu}\|}{2}} \quad \mathcal{N}^{4\sigma^2}_{-\frac{3\|\boldsymbol{\mu}\|}{2}} \quad \mathcal{N}^{4\sigma^2}_{-\frac{3\|\boldsymbol{\mu}\|}{2}} \quad \mathcal{N}^{\sigma^2}_{-\frac{\|\boldsymbol{\mu}\|}{2}} \quad \mathcal{N}^{\sigma^2}_{-\frac{\|\boldsymbol{\mu}\|}{2}} \quad \mathcal{N}^{\sigma^2}_{-\frac{\|\boldsymbol{\mu}\|}{2}} \quad \mathcal{N}^{\sigma^2}_{-\frac{\|\boldsymbol{\mu}\|}{2}} \right] \quad \text{for } i, j \in C_0^2,$$

Next, we will use the following lemma regarding LeakyRelu concentration.

**Lemma 6** (Lemma A.6 in Fountoulakis et al. (2022)). *Fix $s \in \mathbb{N}$, and let $z_1, \ldots, z_s$ be jointly Gaussian random variables with marginals $\boldsymbol{z}_i \sim \mathcal{N}(\mu_i, \sigma_i^2)$. There exists an absolute constant $C > 0$ such that with probability at least $1 - o_s(1)$, we have*

$$\mathrm{LeakyRelu}(z_i) = \mathrm{LeakyRelu}\left(\mu_i\right) \pm C\sigma_i \sqrt{\log s}, \quad \text{for all } i \in [s].$$

Using Lemma 6 with the assumption on $\|\boldsymbol{\mu}\|$ and a union bound, we have that with probability at least $1 - o_n(1)$, LeakyRelu$(\mathbf{\Delta}_{ij})$ is (up to $1 \pm o(1)$)

$$\left[ \frac{\|\boldsymbol{\mu}\|}{2} \quad \frac{-7\beta\|\boldsymbol{\mu}\|}{2} \quad -\frac{3\beta\|\boldsymbol{\mu}\|}{2} \quad -\frac{3\beta\|\boldsymbol{\mu}\|}{2} \quad \frac{\|\boldsymbol{\mu}\|}{2} \quad \frac{\|\boldsymbol{\mu}\|}{2} \quad -\frac{3\beta\|\boldsymbol{\mu}\|}{2} \quad -\frac{3\beta\|\boldsymbol{\mu}\|}{2} \right] \quad \text{for } i, j \in C_1^2,$$

$$\left[ -\frac{7\beta\|\boldsymbol{\mu}\|}{2} \quad \frac{\|\boldsymbol{\mu}\|}{2} \quad -\frac{3\beta\|\boldsymbol{\mu}\|}{2} \quad -\frac{3\beta\|\boldsymbol{\mu}\|}{2} \quad -\frac{3\beta\|\boldsymbol{\mu}\|}{2} \quad -\frac{3\beta\|\boldsymbol{\mu}\|}{2} \quad \frac{\|\boldsymbol{\mu}\|}{2} \quad \frac{\|\boldsymbol{\mu}\|}{2} \right] \quad \text{for } i, j \in C_{-1}^2,$$

$$\left[ -\frac{3\beta\|\boldsymbol{\mu}\|}{2} \quad -\frac{3\beta\|\boldsymbol{\mu}\|}{2} \quad \frac{\|\boldsymbol{\mu}\|}{2} \quad -\frac{7\beta\|\boldsymbol{\mu}\|}{2} \quad -\frac{3\beta\|\boldsymbol{\mu}\|}{2} \quad \frac{\|\boldsymbol{\mu}\|}{2} \quad \frac{\|\boldsymbol{\mu}\|}{2} \quad -\frac{3\beta\|\boldsymbol{\mu}\|}{2} \right] \quad \text{for } i, j \in C_1 \times C_{-1},$$

$$\left[ -\frac{3\beta\|\boldsymbol{\mu}\|}{2} \quad -\frac{3\beta\|\boldsymbol{\mu}\|}{2} \quad -\frac{7\beta\|\boldsymbol{\mu}\|}{2} \quad \frac{\|\boldsymbol{\mu}\|}{2} \quad \frac{\|\boldsymbol{\mu}\|}{2} \quad -\frac{3\beta\|\boldsymbol{\mu}\|}{2} \quad -\frac{3\beta\|\boldsymbol{\mu}\|}{2} \quad \frac{\|\boldsymbol{\mu}\|}{2} \right] \quad \text{for } i, j \in C_{-1} \times C_1,$$

$$\left[ -\frac{\beta\|\boldsymbol{\mu}\|}{2} \quad -\frac{5\beta\|\boldsymbol{\mu}\|}{2} \quad -\frac{5\beta\|\boldsymbol{\mu}\|}{2} \quad -\frac{\beta\|\boldsymbol{\mu}\|}{2} \quad \frac{\|\boldsymbol{\mu}\|}{2} \quad -\frac{\|\boldsymbol{\mu}\|}{2} \quad -\frac{3\beta\|\boldsymbol{\mu}\|}{2} \quad -\frac{\|\beta\boldsymbol{\mu}\|}{2} \right] \quad \text{for } i, j \in C_0 \times C_1,$$

$$\left[ -\frac{5\beta\|\boldsymbol{\mu}\|}{2} \quad -\frac{\beta\|\boldsymbol{\mu}\|}{2} \quad -\frac{\beta\|\boldsymbol{\mu}\|}{2} \quad -\frac{5\beta\|\boldsymbol{\mu}\|}{2} \quad -\frac{3\beta\|\boldsymbol{\mu}\|}{2} \quad -\frac{\beta\|\boldsymbol{\mu}\|}{2} \quad \frac{\|\boldsymbol{\mu}\|}{2} \quad -\frac{\beta\|\boldsymbol{\mu}\|}{2} \right] \quad \text{for } i, j \in C_0 \times C_{-1},$$

$$\left[ -\frac{\beta\|\boldsymbol{\mu}\|}{2} \quad -\frac{5\beta\|\boldsymbol{\mu}\|}{2} \quad -\frac{\beta\|\boldsymbol{\mu}\|}{2} \quad -\frac{5\beta\|\boldsymbol{\mu}\|}{2} \quad -\frac{\beta\|\boldsymbol{\mu}\|}{2} \quad \frac{\|\boldsymbol{\mu}\|}{2} \quad -\frac{\beta\|\boldsymbol{\mu}\|}{2} \quad -\frac{3\beta\|\boldsymbol{\mu}\|}{2} \right] \quad \text{for } i, j \in C_1 \times C_0,$$

$$\left[ -\frac{5\beta\|\boldsymbol{\mu}\|}{2} \quad -\frac{\beta\|\boldsymbol{\mu}\|}{2} \quad -\frac{5\beta\|\boldsymbol{\mu}\|}{2} \quad -\frac{\beta\|\boldsymbol{\mu}\|}{2} \quad -\frac{\beta\|\boldsymbol{\mu}\|}{2} \quad -\frac{3\beta\|\boldsymbol{\mu}\|}{2} \quad -\frac{\|\beta\boldsymbol{\mu}\|}{2} \quad \frac{\|\boldsymbol{\mu}\|}{2} \right] \quad \text{for } i, j \in C_{-1} \times C_0,$$

$$\left[ -\frac{3\beta\|\boldsymbol{\mu}\|}{2} \quad -\frac{3\beta\|\boldsymbol{\mu}\|}{2} \quad -\frac{3\beta\|\boldsymbol{\mu}\|}{2} \quad -\frac{3\beta\|\boldsymbol{\mu}\|}{2} \quad -\frac{\beta\|\boldsymbol{\mu}\|}{2} \quad -\frac{\beta\|\boldsymbol{\mu}\|}{2} \quad -\frac{\beta\|\boldsymbol{\mu}\|}{2} \quad -\frac{\beta\|\boldsymbol{\mu}\|}{2} \right] \quad \text{for } i, j \in C_0^2.$$

Then,

$$
\boldsymbol{r}^T \cdot \mathrm{LeakyRelu}(\boldsymbol{\Delta}_{ij}) = \begin{cases}
10R\beta\|\boldsymbol{\mu}\|(1 \pm o(1)) & \text{if } i,j \in C_1^2 \\
-2R\|\boldsymbol{\mu}\|(1 + 2\beta)(1 \pm o(1)) & \text{if } i,j \in C_{-1}^2 \\
-2R\|\boldsymbol{\mu}\|(1 + 5\beta)(1 \pm o(1)) & \text{if } i \in C_1,\ j \in C_{-1} \\
10R\beta\|\boldsymbol{\mu}\|(1 \pm o(1)) & \text{if } i \in C_{-1},\ j \in C_1 \\
-\frac{R}{2}\|\boldsymbol{\mu}\|(1 - 21\beta)(1 \pm o(1)) & \text{if } i \in C_0,\ j \in C_1 \\
-\frac{R}{2}\|\boldsymbol{\mu}\|(1 - 11\beta)(1 \pm o(1)) & \text{if } i \in C_0,\ j \in C_{-1} \\
-\frac{R}{2}\|\boldsymbol{\mu}\|(1 - 5\beta)(1 \pm o(1)) & \text{if } i \in C_1,\ j \in C_0 \\
-\frac{R}{2}\|\boldsymbol{\mu}\|(1 - 5\beta)(1 \pm o(1)) & \text{if } i \in C_{-1},\ j \in C_0 \\
2R\beta\|\boldsymbol{\mu}\|(1 \pm o(1)) & \text{if } i,j \in C_0^2
\end{cases},
$$

and the proof is complete. $\qquad\square$

Next we will define our high probability event.

**Definition 2.** $\boldsymbol{\mathcal{E}}' \overset{\text{def}}{=} \boldsymbol{\mathcal{E}} \cap \boldsymbol{\mathcal{E}}^*$, where $\boldsymbol{\mathcal{E}}^*$ is the event that for a fixed $\boldsymbol{w} \in \mathbb{R}^d$, all $i \in [n]$ satisfy $|\boldsymbol{w}^T \mathbf{X}_i - \mathbf{E}[\boldsymbol{w}^T \mathbf{X}_i]| \leq 10\sigma\|\boldsymbol{w}\|_2 \sqrt{\log n}$.

The following lemma is obtained by using Lemma 4 with standard Gaussian concentration and a union bound.

**Lemma 7.** With probability at least $1 - 1/\mathrm{poly}(n)$ event $\boldsymbol{\mathcal{E}}'$ holds.

**Corollary 8.** Suppose that $p, q$ satisfy Assumption 1, $\|\boldsymbol{\mu}\| = \omega(\sigma\sqrt{\log n})$ and fix $R \in \mathbb{R}$. Then, there exists a choice of attention architecture $\Psi$ such that with probability $1 - o_n(1)$ over $(\mathbf{A}, \mathbf{X}) \sim \mathsf{CSBM}(n, p, q, \boldsymbol{\mu}, \sigma^2)$ it holds that

$$
\gamma_{ij} = \begin{cases}
\frac{3}{np}(1 \pm o(1)) & \text{if } i,j \in C_0^2 \cup C_1^2 \\
\frac{3}{nq}(1 \pm o(1)) & \text{if } i,j \in C_{-1} \times C_1 \\
\frac{3}{nq}\exp(-\Theta(R\|\boldsymbol{\mu}\|)) & \text{if } i,j \in C_{-1} \times C_{-1} \cup C_0 \\
\frac{3}{np}\exp(-\Theta(R\|\boldsymbol{\mu}\|)) & \text{otherwise}
\end{cases},
$$

where $R$ is a parameter of the architecture.

**Proof.** The proof is immediate. First applying the ansatz from Lemma 5 with $\beta < 1/25$, Lemma 7 and a union bound. Using the definition of $\gamma_{ij}$ concludes the proof. $\qquad\square$

Next, we prove Thm. 1 that the model distinguish nodes from $C_0$ for any choice of $p, q$ satisfying Assumption 1. We restate the theorem for convince.

**Theorem 9** (Formal restatement of Thm. 1). Suppose that $p, q$ satisfy Assumption 1 and $\|\boldsymbol{\mu}\|_2 = \omega(\sigma\sqrt{\log n})$. Then, there exists a choice of attention architecture $\Psi$ such that with probability at least $1 - o_n(1)$ over the data $(\mathbf{X}, \mathbf{A}) \sim \mathsf{CSBM}(n, p, q, \boldsymbol{\mu}, \sigma^2)$, the estimator

$$
\hat{x}_i \overset{\text{def}}{=} \sum_{j \in N_i} \gamma_{ij}\tilde{\boldsymbol{w}}^T \mathbf{X}_j + b \ \text{ where } \tilde{\boldsymbol{w}} = \boldsymbol{\mu}/\|\boldsymbol{\mu}\|,\ b = -\|\boldsymbol{\mu}\|/2
$$

satisfies $\hat{x}_i < 0$ if and only if $i \in C_0$.

**Proof.** Let $\Psi$ be the architecture from Cor. 8 and let $R$ satisfy $R\|\boldsymbol{\mu}\|_2 = \omega(1)$. We will compute the mean and variance of the estimator $\hat{x}_i$ conditioned on $\boldsymbol{\mathcal{E}}'$. Suppose that $i \in C_0$. By using Cor. 8, Definition 2 and our assumption on $\|\boldsymbol{\mu}\|$ and $R$, we have

$$
\max\left\{\frac{3}{np}\exp(-\Theta(R\|\boldsymbol{\mu}\|)), \frac{3}{nq}\exp(-\Theta(R\|\boldsymbol{\mu}\|))\right\} = o\left(\frac{1}{n(p + 2q)}\right),
$$

and therefore

$$
\begin{aligned}
\mathbf{E}\left[\hat{x}_i \mid \boldsymbol{\mathcal{E}}'\right] &= \mathbf{E}\left[\sum_{k \in \{-1,0,1\}} \sum_{j \in N_i \cap C_k} \gamma_{ij} \tilde{\boldsymbol{w}}^T \mathbf{X}_j \mid \boldsymbol{\mathcal{E}}'\right] - \frac{\|\boldsymbol{\mu}\|}{2} \\
&= \mathbf{E}[|C_0 \cap N_i| \mid \boldsymbol{\mathcal{E}}']\left(\pm \frac{3}{np}(1 \pm o(1)) \cdot 10\sigma\sqrt{\log n}\right) \\
&\quad + \mathbf{E}[|C_1 \cap N_i| \mid \boldsymbol{\mathcal{E}}']\left(o\left(\frac{1}{n(p+2q)}\right) \cdot (\|\boldsymbol{\mu}\| \pm 10\sigma\sqrt{\log n})\right) \\
&\quad + \mathbf{E}[|C_{-1} \cap N_i| \mid \boldsymbol{\mathcal{E}}']\left(o\left(\frac{1}{n(p+2q)}\right) \cdot (-\|\boldsymbol{\mu}\| \pm 10\sigma\sqrt{\log n})\right) - \frac{\|\boldsymbol{\mu}\|}{2} \\
&= -\frac{\|\boldsymbol{\mu}\|}{2}(1 \pm o(1)).
\end{aligned}
$$

By similar reasoning we have that for $i \in C_{-1} \cup C_1$, $\mathbf{E}\left[\hat{x}_i \mid \boldsymbol{\mathcal{E}}'\right] = \frac{\|\boldsymbol{\mu}\|}{2}(1 \pm o(1))$.

Next, we claim that for each $i \in [n]$ the random variable $\hat{x}_i$ given $\boldsymbol{\mathcal{E}}'$ is sub-Gaussian with a small sub-Gaussian constant compared to the above expectation. The following lemma is a straightforward adaptation of Lemma A.11 in Fountoulakis et al. (2022), and we provide its proof for completeness.

**Lemma 10.** Conditioned on $\boldsymbol{\mathcal{E}}'$, the random variables $\{\hat{\boldsymbol{x}}_i\}_i$ are sub-Gaussian with parameter $\tilde{\sigma}_i^2 = O\left(\frac{\sigma^2}{np}\right)$ if $i \in C_0 \cup C_1$ and $\tilde{\sigma}_i^2 = O\left(\frac{\sigma^2}{nq}\right)$ otherwise.

**Proof.** Fix $i \in [n]$, and write $\mathbf{X}_i = \varepsilon_i \boldsymbol{\mu} + \sigma \boldsymbol{g}_i$ where $\boldsymbol{g}_i \sim \mathcal{N}(0, \mathbf{I}_d)$, and $\varepsilon_i$ denotes the class membership. Consider $\hat{x}_i$ as a function of $\boldsymbol{g} = [\boldsymbol{g}_1 \circ \boldsymbol{g}_2 \circ \cdots \circ \boldsymbol{g}_n] \in \mathbb{R}^{nd}$, where $\circ$ denotes vertical concatenation. Namely, consider the function

$$
\hat{x}_i = f_i(\boldsymbol{g}) \overset{\text{def}}{=} \sum_{j \in N_i} \gamma_{ij}(\boldsymbol{g}) \, \tilde{\boldsymbol{w}}^T(\varepsilon_j \boldsymbol{\mu} + \sigma \boldsymbol{g}_j) - \|\boldsymbol{\mu}\|/2, \quad i \in [n].
$$

Since $\boldsymbol{g} \sim \mathcal{N}(0, \mathbf{I}_{nd})$, proving that $\hat{x}_i$ given $\boldsymbol{\mathcal{E}}'$ is sub-Gaussian for each $i \in [n]$, reduces to showing that the function $f_i : \mathbb{R}^{nd} \to \mathbb{R}$ is Lipschitz over $E \subseteq \mathbb{R}^{nd}$ defined by $\boldsymbol{\mathcal{E}}'$ and the relation $\mathbf{X}_i = \varepsilon_i \boldsymbol{\mu} + \sigma \boldsymbol{g}_i$. That is, $E \overset{\text{def}}{=} \left\{\boldsymbol{g} \in \mathbb{R}^{nd} \mid |\tilde{\boldsymbol{w}}^T \boldsymbol{g}_i| \leq 10\sqrt{\log n}, \forall i \in [n]\right\}$. Specifically, we show that conditioning on the event $\boldsymbol{\mathcal{E}}'$ (which restricts $\boldsymbol{g} \in E$), the Lipschitz constant $L_{f_i}$ of $f_i$ satisfies $L_{f_i} = O\left(\frac{\sigma}{\sqrt{np}}\right)$ for $i \in C_0 \cup C_1$ and $L_{f_i} = O\left(\frac{\sigma}{nq}\right)$ otherwise, and hence proving the claim.

First note that event $\boldsymbol{\mathcal{E}}'$ induce a transformation which transforms isotropic Gaussians to truncated Gaussians vectors. Similarly to Fountoulakis et al. (2022), we can show that this transformation can be obtained by a push-forward mapping whose Lipschitz constant is 1.

$$
\bar{\boldsymbol{v}} = M(\boldsymbol{v}) \overset{\text{def}}{=} [\tau(\boldsymbol{v}_1), \tau(\boldsymbol{v}_2), \ldots, \tau(\boldsymbol{v}_n)]^T \tag{8}
$$

where $\tau(x) \overset{\text{def}}{=} \Phi^{-1}((1 - 2c)\Phi(x) + c)$ for $c = \Phi(-10\sqrt{\log n})$.

To compute the Lipschitz constant of $f_i(\boldsymbol{g})$ for $i \in [n]$, let us denote $\mathbf{X} = [\mathbf{X}_1 \circ \mathbf{X}_2 \circ \cdots \circ \mathbf{X}_n]$ and consider the function

$$
\tilde{f}_i(\mathbf{X}) \overset{\text{def}}{=} \sum_{j \in N_i} \gamma_{ij}(\mathbf{X}) \, \tilde{\boldsymbol{w}}^T \mathbf{X}_j, \quad i \in [n]
$$

Let us assume without loss of generality that $i \in C_0$ (the cases for $i \in C_1$ and $i \in C_{-1}$ are obtained identically). Conditioning on the event $\boldsymbol{\mathcal{E}}'$, which imposes the restriction that $\mathbf{X} \in \tilde{E}$ where

$$
\tilde{E} \overset{\text{def}}{=} \left\{\mathbf{X} \in \mathbb{R}^{nd} \mid |\mathbf{X}_i - \varepsilon_i \boldsymbol{\mu}| \leq 10\sigma\sqrt{\log n}, \forall i \in [n]\right\}.
$$

Conditioning on $\mathcal{E}'$ (which restricts $\mathbf{X}, \mathbf{X}' \in \tilde{E}$), using Cor. 8 and recalling that $R$ satisfies $R\|\boldsymbol{\mu}\|_2 = \omega(1)$, we get[3]

$$\left| \tilde{f}_i(\mathbf{X}) - \tilde{f}_i(\mathbf{X}') \right|$$

$$\simeq \left| \sum_{j \in N_i \cap C_0} \frac{3}{np} \tilde{\boldsymbol{w}}^T (\mathbf{X}_j - \mathbf{X}'_j) + \sum_{j \in N_i \cap C_1} \frac{3}{np} \cdot e^{-\Theta(R\|\boldsymbol{\mu}\|_2)} \tilde{\boldsymbol{w}}^T (\mathbf{X}_j - \mathbf{X}'_j) + \sum_{j \in N_i \cap C_{-1}} \frac{3}{np} \cdot e^{-\Theta(R\|\boldsymbol{\mu}\|_2)} \tilde{\boldsymbol{w}}^T (\mathbf{X}_j - \mathbf{X}'_j) \right|$$

$$= \left| \begin{bmatrix} \frac{3}{np}(1 \pm o(1))\tilde{\boldsymbol{w}} & \text{if } j \in N_i \cap C_0 \\ \frac{3}{np} \exp(-\Theta(R\|\boldsymbol{\mu}\|_2))(1 \pm o(1))\tilde{\boldsymbol{w}} & \text{if } j \in N_i \cap C_1 \\ \frac{3}{np} \exp(-\Theta(R\|\boldsymbol{\mu}\|_2))(1 \pm o(1))\tilde{\boldsymbol{w}} & \text{if } j \in N_i \cap C_{-1} \\ 0 & \text{if } j \notin N_i \end{bmatrix}_{j \in [n]}^T (\mathbf{X} - \mathbf{X}') \right|$$

$$\leq \left\| \begin{bmatrix} \frac{3}{np}(1 \pm o(1))\tilde{\boldsymbol{w}} & \text{if } j \in N_i \cap C_0 \\ \frac{3}{np} \exp(-\Theta(R\|\boldsymbol{\mu}\|_2))(1 \pm o(1))\tilde{\boldsymbol{w}} & \text{if } j \in N_i \cap C_1 \\ \frac{3}{np} \exp(-\Theta(R\|\boldsymbol{\mu}\|_2))(1 \pm o(1))\tilde{\boldsymbol{w}} & \text{if } j \in N_i \cap C_{-1} \\ 0 & \text{if } j \notin N_i \end{bmatrix}_{j \in [n]} \right\|_2 \|\mathbf{X} - \mathbf{X}'\|_2$$

$$\leq \sqrt{\frac{3}{np}} (1 + o(1)) \|\tilde{\boldsymbol{w}}\|_2 \|\mathbf{X} - \mathbf{X}'\|_2$$

$$= \sqrt{\frac{3}{np}} (1 + o(1)) \|\mathbf{X} - \mathbf{X}'\|_2 .$$

This shows the Lipschitz constant of $\tilde{f}_i(\mathbf{X})$ over $\tilde{E}$ satisfies $L_{\tilde{f}_i} = O\left(\frac{1}{\sqrt{np}}\right)$. On the other hand, by viewing $\mathbf{X}$ as a function of $\boldsymbol{g}$, it is straightforward to see that the function $h(\boldsymbol{g}) : \mathbb{R}^{nd} \to \mathbb{R}^{nd}$ defined by $h(\boldsymbol{g}) \stackrel{\text{def}}{=} \mathbf{X}(\boldsymbol{g})$ has Lipschitz constant $L_h = \sigma$, as

$$\|h(\boldsymbol{g}) - h(\boldsymbol{g}')\|_2 = \|\boldsymbol{\varepsilon}\boldsymbol{\mu} + \sigma\boldsymbol{g} - (\boldsymbol{\varepsilon}\boldsymbol{\mu} + \sigma\boldsymbol{g}')\|_2 = \sigma \|\boldsymbol{g} - \boldsymbol{g}'\|_2.$$

Therefore, since $f_i(\boldsymbol{g}) = \tilde{f}_i(h(\boldsymbol{g}))$ and $\boldsymbol{g} \in E$ if and only if $\mathbf{X} \in \tilde{E}$, we have that, conditioning on $\mathcal{E}'$, the function $\hat{x}_i = f_i(\boldsymbol{g})$ is Lipschitz continuous with Lipschitz constant $L_{f_i} = L_{\tilde{f}_i} L_h = O\left(\frac{\sigma}{\sqrt{np}}\right)$. Since $\boldsymbol{g} \sim \mathcal{N}(0, \mathbf{I}_{nd})$, we know that $\hat{x}_i$ is sub-Gaussian with sub-Gaussian constant $\tilde{\sigma}^2 = L_{f_i}^2 = O\left(\frac{\sigma^2}{np}\right)$. $\qquad\square$

The following lemma will be used for bounding the misclassification probability.

**Lemma 11** (Rigollet & Hütter (2015))**.** Let $x_1, \ldots, x_n$ be sub-Gaussian random variables with the same mean and sub-Gaussian parameter $\tilde{\sigma}^2$. Then,

$$\mathbf{E}\left[\max_{i \in [n]} (x_i - \mathbf{E}[x_i])\right] \leq \tilde{\sigma}\sqrt{2 \log n}.$$

Moreover, for any $t > 0$

$$\mathbf{Pr}\left[\max_{i \in [n]} (x_i - \mathbf{E}[x_i]) > t\right] \leq 2n \exp\left(-\frac{t^2}{2\tilde{\sigma}^2}\right).$$

We bound the probability of misclassification

$$\mathbf{Pr}\left[\max_{i \in C_0} \hat{x}_i \geq 0\right] \leq \mathbf{Pr}\left[\max_{i \in C_0} \hat{x}_i > t + \mathbf{E}[\hat{x}_i]\right],$$

for $t < |\mathbf{E}[\hat{x}_i]| = \frac{\|\boldsymbol{\mu}\|_2}{2}(1 \pm o(1))$. By Lemma 10, picking $t = \Theta\left(\sigma\sqrt{\log |C_0|}\right)$ and applying Lemma 11 implies that the above probability is $1/\text{poly}(n)$.

---

[3]We drop the $(1 \pm o(1))$ in the first line of the computation for compactness and use $\simeq$ as notation.

Similarly for class $C_1 \cup C_{-1}$ we have that the misclassification probability is

$$\mathbf{Pr}\left[\min_{i \in C_1 \cup C_{-1}} \hat{x}_i \le 0\right] = \mathbf{Pr}\left[-\max_{i \in C_1 \cup C_{-1}} (-\hat{x}_i) \le 0\right] = \mathbf{Pr}\left[\max_{i \in C_1 \cup C_{-1}} (-\hat{x}_i) \ge 0\right]$$
$$\le \mathbf{Pr}\left[\max_{i \in C_1 \cup C_{-1}} -\hat{x}_i > t - \mathbf{E}[\hat{x}_i]\right],$$

for $t < \mathbf{E}[\hat{x}_i]$. Picking $t = \Theta\left(\sigma\sqrt{\log|C_1 \cup C_{-1}|}\right)$ and applying Lemma 11 and a union bound over the misclassification probabilities of both classes conclude the proof of the corollary. $\square$

Combining Thm. 9 with Lemma 3, we immediately get Cor. 2 which we restate below.

**Corollary 12.** Suppose $p, q = \Omega(\log^2 n/n)$ and $\|\boldsymbol{\mu}\| \ge \omega\left(\sigma\sqrt{\frac{(p+2q)\log n}{n(p-q)^2}}\right)$. Then, there is a choice of attention architecture $\Psi$ such that, with probability at least $1 - o(1)$ over the data $(\mathbf{X}, \mathbf{A}) \sim \mathsf{CSBM}(n, p, q, \boldsymbol{\mu}, \sigma^2)$, CAT separates nodes $C_0$ from $C_1 \cup C_{-1}$.

## B  SYNTHETIC EXPERIMENTS

In this section, we present the complete results for the synthetic data experiments of §6.1. First, we describe the parameterization we use for the 1-layer GCN, GAT, and CAT models; then, we verify the behavior of the normalized score function ($\gamma_{ij}$) matches that of the theory presented in Cor. 8. In particular, we visualize the average of the following three groups of gammas (Fig. 4):

- Gammas $\gamma_{ij}$ included in $i, j \in C_0^2 \cup C_1^2$. Solid lines.
- Gammas $\gamma_{ij}$ included in $i, j \in C_{-1} \times C_1$. Dashed lines.
- The rest of gammas. Dotted lines.

For completeness, we also include the empirical results that validate Thm. 1 and Cor. 2, which were discussed already in §6.1.

**Experimental setup.** We assume the following parametrization for the 1-layer GCN, GAT, and CAT:

$$h_i' = \left( \sum_{j \in N_i} \gamma_{ij} \tilde{w}^T \mathbf{X}_j \right) - C \cdot \|\boldsymbol{\mu}\|/2 \, , \tag{9}$$

where $N_i$ are the set of neighbors of node $i$, $\mathbf{X}_j$ are the features of node $j$—obtained from the CSBM described in §4, and $h_i'$ are the logits of the prediction of node $i$. Note that for GCN we have $\gamma_{ij} = \frac{1}{|N_i^*|}$. Otherwise, we consider the following parameterization of the score function $\Psi$:

$$\gamma_{ij} = \frac{\exp\left(\Psi(h_i, h_j)\right)}{\sum_{k \in N_i^*} \exp(\Psi(h_i, h_k))} \quad \text{where} \tag{10}$$

$$\Psi(h_i, h_j) \overset{\text{def}}{=} r^T \cdot \text{LeakyRelu}\left( \mathbf{S} \cdot \begin{bmatrix} \tilde{w}^T h_i \\ \tilde{w}^T h_j \end{bmatrix} + b \right) \, . \tag{11}$$

For these experiments, we define the parameters $\tilde{w}$, $\mathbf{S}$, $b$ and $r$ as in the proofs in App. A:

$$\tilde{w} \overset{\text{def}}{=} \frac{\boldsymbol{\mu}}{\|\boldsymbol{\mu}\|}, \qquad \mathbf{S} \overset{\text{def}}{=} \begin{bmatrix} 1 & 1 \\ -1 & -1 \\ 1 & -1 \\ -1 & 1 \\ 0 & 1 \\ 1 & 0 \\ 0 & -1 \\ -1 & 0 \end{bmatrix}, \qquad b \overset{\text{def}}{=} \begin{bmatrix} -3/2 \\ -3/2 \\ -3/2 \\ -3/2 \\ -1/2 \\ -1/2 \\ -1/2 \\ -1/2 \end{bmatrix} \cdot \|\boldsymbol{\mu}\| \cdot C, \qquad r \overset{\text{def}}{=} R \cdot \begin{bmatrix} 2 \\ -2 \\ -2 \\ 2 \\ -1 \\ -1 \\ -1 \\ -1 \end{bmatrix}, \tag{12}$$

where $R > 0$ and $C > 0$ are arbitrary scaling parameters. Both $C$ and $R$ and input to the score function are set different for each of the models, as indicated in Table 4. In particular, we set $R = \frac{7}{\|\boldsymbol{\mu}\|}$ for both GAT and CAT such that: i) all $\gamma_{ij}$ are distinguishable as we decrease $\|\boldsymbol{\mu}\|$; and ii) we avoid numerical instabilities in the implementation when computing the exponential of $R \times \|\boldsymbol{\mu}\|$ in order to obtain $\gamma_{ij}$ (see Cor. 8), as the exponential of small or large values leads to under/overflow issues. As for $C$, we set $C = 1$ for GAT and $C = (p - q)/(p + 2q)$ for CAT such that we counteract the fact that the distance between classes shrink as we increase $q$, see Lemma 3:

Regarding the data model, we set (as described in §6.1) $n = 10000$, $p = 0.5$, $\sigma = 0.1$, and $d = n/\left(5 \log^2(n)\right)$. We set the slope of the LeakyReLU activation to $\beta = 1/5$ for the GAT and $\beta = 0.01$ for CAT, such that the proof of Cor. 8 is valid. As described in the main paper, to assess the sensitivity to structural noise, we present the complete results for two sets of experiments. First, we vary the noise level $q$ between 0 and 0.5, fixing the mean vector $\boldsymbol{\mu}$. We test two values of $\|\boldsymbol{\mu}\|$: the first corresponds to the *easy* regime ($\|\boldsymbol{\mu}\| = 10\sigma\sqrt{2 \log n} \approx 4.3$) where classes are far apart; and the second correspond to the *hard* regime ($\|\boldsymbol{\mu}\| = \sigma = 0.1$)

Table 4: Parameters for the synthetic experiments.

| Model | $C$ | $R$ | $\boldsymbol{h}_i$ |
|-------|-----|-----|--------------------|
| GCN | 0 | $-$ | $-$ |
| GAT | 1 | $\frac{7}{\|\boldsymbol{\mu}\|}$ | $\mathbf{X}_i$ |
| CAT | $\frac{p-q}{p+2q}$ | $\frac{7}{\|\boldsymbol{\mu}\|}$ | $\frac{1}{|N_i^*|}\sum_{k\in N_i^*}\mathbf{X}_k$ |

Figure 4: Synthetic data results. On the top row, we show the node classification, and in the following two rows we show the $\gamma_{ij}$ values for GAT and CAT respectively. In the two left-most figures, we show how the results vary with the noise level $q$ for $\|\boldsymbol{\mu}\| = 0.1$ and $\|\boldsymbol{\mu}\| = 4.3$. In the two right-most figures, we show how the results vary with the norm of the means $\|\boldsymbol{\mu}\|$ for $q = 0.1$ and $q = 0.3$. We use two vertical lines to present the classification threshold stated in Thm. 1 (solid line) and Cor. 2 (dashed line).

where the distance between the clusters is small. In the second experiment we modify instead the distance between the means, sweeping $\|\boldsymbol{\mu}\|$ in the range $\left[\sigma/20, 20\sigma\sqrt{2\log n}\right]$ which corresponds to $[0.005, 8.58]$, and thus covering the transition from the hard setting (small $\|\boldsymbol{\mu}\|$) to the easy one (large $\|\boldsymbol{\mu}\|$). In these experiments, we fix $q$ to 0.1 (low noise) and 0.3 (high noise).

**Results** are summarized in Fig. 4. The top row contains the node classification performance for each of the models (i.e., Fig. 1), the next two rows contain the $\gamma_{ij}$ values for GAT and CAT respectively. The two left-most columns of Fig. 4 show the results for the hard and easy regimes, respectively, as we vary the noise level $q$. In the hard regime, we observe that GAT is unable to achieve separation for any value of $q$, whereas CAT achieves perfect classification when $q$ is small enough. The gamma plots help shed some light on this question. For GAT, we observe that the gammas represented with the dotted and solid lines collapse for any value of $q$ (see middle plot), while this does not happen for CAT when the noise level is low (see bottom plot). This exemplifies the advantage of CAT over GAT as stated in Cor. 2. When the distance between the means is large enough, we see that GAT achieves perfect results independently of $q$, as stated in Thm. 1. We also observe that, in this case,

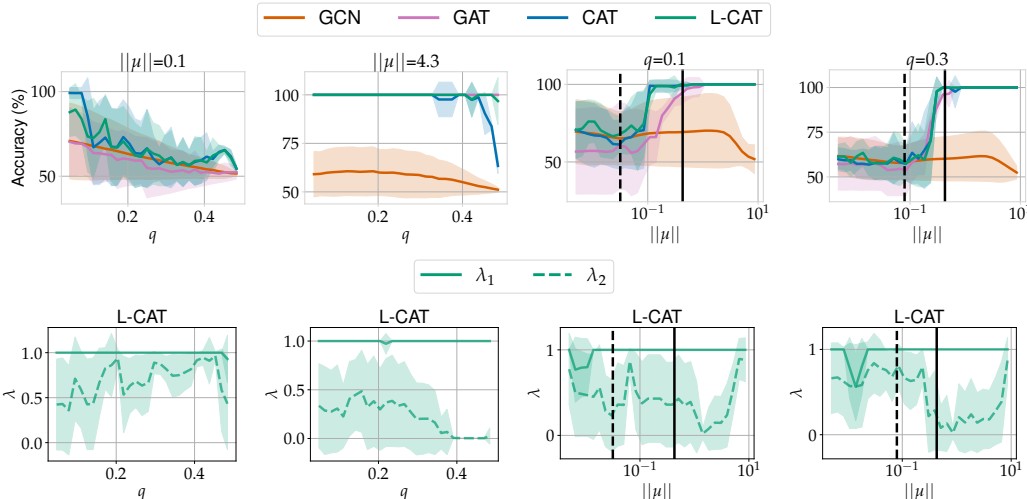

Figure 5: Synthetic data results learning $C$, $\lambda_1$ and $\lambda_2$. On the top row, we show the node classification accuracy, and in the bottom row we show the learned values of $\lambda_1$ and $\lambda_2$ for L-CAT. In the two left-most figures, we show how the results vary with the noise level $q$ for $\|\boldsymbol{\mu}\| = 0.1$ and $\|\boldsymbol{\mu}\| = 4.3$. In the two right-most figures, we show how the results vary with the norm of the means $\|\boldsymbol{\mu}\|$ for $q = 0.1$ and $q = 0.3$. We use two vertical lines to present the classification threshold stated in Thm. 1 (solid line) and Cor. 2 (dashed line).

the gammas represented with the dotted and solid lines do not collapse for any value of $q$. In contrast, as we increase $q$, CAT fails to satisfy the condition in Cor. 2, and therefore achieves inferior performance. We note that the low performance is due to the fact that all gammas collapse to the same value for large noise levels.

For the second set of experiments (two right-most columns of Fig. 1), where we fix $q$ and sweep $\|\boldsymbol{\mu}\|$, we observe that, for both values of $q$, there exists a transition in the accuracy of both GAT and CAT as a function of $\|\boldsymbol{\mu}\|$. As shown in the main manuscript, GAT achieves perfect accuracy when the distance between the means satisfies the condition in Thm. 1 (solid vertical line in Fig. 1). Moreover, we can see the improvement CAT obtains over GAT. Indeed, when $\|\boldsymbol{\mu}\|$ satisfies the conditions of Cor. 2 (dashed vertical line in Fig. 1), the classification threshold is improved. As we increase $q$, we see that the gap between the two vertical lines decreases, which means that the improvement decreases as $q$ increments, exactly as stated in Cor. 2. This transition from the hard regime to the easy regime is also observed in the gamma plots: we observe the largest difference in value between the different groups of lambdas for values of $\|\boldsymbol{\mu}\|$ that satisfy the condition in Thm. 1 (that is to the right of the vertical lines).

### B.1   OTHER EXPERIMENTS

In the following, we extend the results for the synthetic data presented above. In particular, we aim to evaluate if L-CAT is able to achieve top performance regardless of the scenario. That is, we want to evaluate if L-CAT consistently performs at least as good as the best-performing model. We change the fixed-parameter setting of the previous section and, instead, we evaluate the performance of GCN, GAT, CAT and L-CAT when we learn the model-dependent parameters.

**Experimental setup.** We assume the same parametrization for the 1-layer GCN, GAT and CAT described in Eq. 9 and Eq. 11. For L-CAT, we add the parameters $\lambda_1$ and $\lambda_2$, as indicated in Eq. 7. We fix the parameters shared among the models, that is, $\tilde{\boldsymbol{w}}$, $\mathbf{S}$, $\boldsymbol{b}$, $\boldsymbol{r}$, and $R$, with the values indicated in Eq. 12. Different from previous experiments, we now learn $C$ and, for L-CAT, we also learn $\lambda_1$ and $\lambda_2$. We choose to fix part of the parameters (instead of learning them all) to keep the problem as similar as possible to the theoretical analysis we provided in §4 and App. A. If we instead learn all the parameters, it takes a single dimension

of the features to be (close to) linearly separable to find a solution that achieves a similar performance regardless of the model, which hinders the analysis. This is a consequence of the probabilistic nature of the features. One way of solving this issue would be to make $n$ big enough. Instead, we opt to have a fixed $n$ and reduce the degrees of freedom of the models by fixing the parameters shared across all models. The rest of the experimental setup matches the one from App. B. Additionally, we use the Adam optimizer (Kingma & Ba, 2015) with a learning rate of 0.05, and we train for 100.

**Results** are summarized in Fig. 5. The top row contains the node classification performance for every model, while the bottom row contains the learned values of $\lambda_1$ (solid line) and $\lambda_2$ (dashed line) with L-CAT. The two left-most columns of Fig. 5 show the results for the hard and easy regimes, respectively, as we vary the noise level $q$. In the hard regime, we see rather noisy results. Still, the behaviour is similar to that of Fig. 4: the performance of CAT degrades as we increase $q$. We also observe that, on average, CAT outperforms GAT. In this case, we observe that L-CAT achieves similar performance as CAT, which can be explained by inspecting the learned values of lambda in the bottom row. We observe that $\lambda_1 = 1$ and $\lambda_2 \geq 0.5$ on average for all values of $q$. This indicates that L-CAT is closer to CAT than to GAT. When the distance between the means is large enough (i.e., $\|\boldsymbol{\mu}\| = 4.3$), we see that GAT achieves perfect results independently of $q$ while the performance of CAT deteriorates with large values of $q$, the same trend as in Fig. 4. Remarkably, we observe that L-CAT also achieves perfect results independently of $q$. If we inspect the lambda values, we first see that $\lambda = 1$ for all $q$, thus the interpolation happens between CAT and GAT. Looking at the values of $\lambda_2$, we observe that, for small values of $q$, $\lambda_2$ is pretty noisy, which is expected since any solution achieves perfect performance. Interestingly, we have that $\lambda_2 = 0$ for large values of $q$, with negligible variance. This indicates that L-CAT learns that it must behave like GAT in order to perform well.

For the second set of experiments (two right-most columns of Fig. 5), we fix $q$ and sweep $\|\boldsymbol{\mu}\|$ like we did in Fig. 4. Here, we observe a similar trend: for both values of $q$, there exists a transition in the accuracy of both GAT and CAT as a function of $\|\boldsymbol{\mu}\|$. Yet once again, we observe that L-CAT consistently achieves a similar performance to the best-performing model in every situation.

## C  DATASET DESCRIPTION

We present further details about the datasets used in our experiments, summarized in Table 5. All datasets are undirected (or transformed to undirected otherwise) and transductive.

The upper rows of the table refer to datasets used in §6.2 taken from the PyTorch Geometric framework.[4] The following paragraphs present a short description of such datasets.

**Amazon Computers & Photos** are datasets taken from Shchur et al. (2018), in which nodes represent products, and edges indicate that the products are usually bought together. The node features are a Bag of Words (BoW) representation of the product reviews. The task is to predict the category of the products.

**GitHub** is a dataset introduced in Rozemberczki et al. (2021), in which nodes correspond to developers, and edges indicate mutual follow relationship. Node features are embeddings extracted from the developer's starred repositories and profile information (e.g., location or employer). The task is to infer whether a node relates to web or machine learning development.

**FacebookPagePage** is a dataset introduced in Rozemberczki et al. (2021), where nodes are Facebook pages, and edges imply mutual likes between the pages. Nodes features are text embeddings extracted from the pages' description. The task consist on identifying the page's category.

**TwitchEN** is a dataset introduced in Rozemberczki et al. (2021). Here, nodes correspond to Twitch gamers, and links reveal mutual friendship. Node features are an embedding of

---

[4]https://pytorch-geometric.readthedocs.io/en/latest/modules/datasets.html

games liked, location, and streaming habits. The task is to infer if a gamer uses explicit content.

**Coauthor Physics & CS** are datasets introduced in Shchur et al. (2018). In this case, nodes represent authors which are connected with an edge if they have co-authored a paper. Node features are BoW representations of the keywords of the author's papers. The task consist on mapping each author to their corresponding field of study.

**DBLP** is a dataset introduced in Bojchevski & Günnemann (2018) that represents a citation network. In this dataset, nodes represent papers and edges correspond to citations. Node features are BoW representations of the keywords of the papers. The task is to predict the research area of the papers.

**PubMed, Cora & CiteSeer** are citation networks introduced in Yang et al. (2016). Nodes represent documents, and edges refer to citations between the documents. Node features are BoW representations of the documents. The task is to infer the topic of the documents.

The bottom rows of Table 5 refer to the datasets from Open Graph Benchmark (OGB) (Hu et al., 2020a) [5] used in §6.3. We include a short description of them in the paragraphs below.

**ogbn-arxiv** is a citation network of computer science papers in arXiv (Wang et al., 2020). Nodes represent papers, and directed edges refer to citations among them. Node features are embeddings of the title and abstract of the papers. The task is to predict the research area of the nodes.

**ogbn-products** contains a co-purchasing network (Bhatia et al., 2016). Nodes represent products, and links are present whenever two products are bought together. Node features are embeddings of a BoW representation of the product description. The task is to infer the category of the products.

**ogbn-mag** is a heterogeneous network formed from a subgraph of the Microsoft Academic Graph (MAG) (Wang et al., 2020). Nodes can belong to one of these four types: authors, papers, institutions and fields of study. Moreover, directed edges belong to one of the following categories: "author is affiliated with an institution," "author has written a paper," "paper cites a paper," and "paper belongs to a research area." Only nodes that are papers contain node features, which are a text embedding of the document content. The task is to predict the venue of the nodes that are papers.

**ogbn-proteins** is a network whose nodes represent proteins and edges indicate different types of associations among them. This dataset does not contain node features. The tasks are to predict multiple protein functions, each of them being a binary classification problem.

## D  REAL DATA EXPERIMENTS

### D.1  EXPERIMENTAL DETAILS

**Computational resources.** We used CPU cores to run this set of experiments. In particular, for each trial, we used 2 CPU cores and up to 16 GB of memory. We ran the experiments in parallel using a shared cluster with 10000 CPU cores approximately.

**General experimental setup.** As mentioned in §6.2, we repeat all experiments 10 times, which correspond to 10 different random initialization of the parameters of the GNNs. In all cases, we choose the model parameters with the best validation performance during training. In order to run the experiments and collect the results, we used the GraphGym framework (You et al., 2020), which includes the data processing and loading of the datasets, as well as the evaluation and collection of the results. We split the datasets in 70 % training, 15 % validation, and 15 % test.

We cross-validate the number of message-passing layers in the network $(2, 3, 4)$, as well as the learning rate $([0.01, 0.005])$. Then, we report the results of the best validation error among the 4 possible combinations. However, in practice we found the best performance always to

---

[5] https://ogb.stanford.edu/docs/nodeprop

Table 5: Dataset statistics. On the top part of the table, we show the datasets used in §6.2. On the bottom part of the table, we show the datasets used in §6.3.

| Name | #Nodes | #Edges | Avg. degree | #Node feats. | #Edge feats. | #Tasks | Task Type |
|------|--------|--------|-------------|--------------|--------------|--------|-----------|
| AmazonComp. | 13,752 | 491,722 | 35.76 | 767 | - | 1 | 10-class clf. |
| AmazonPhoto | 7,650 | 238,162 | 31.13 | 745 | - | 1 | 8-class clf. |
| GitHub | 37,700 | 578,006 | 15.33 | 128 | - | 1 | Binary clf. |
| FacebookP. | 22,470 | 342,004 | 15.22 | 128 | - | 1 | 4-class clf. |
| CoauthorPh. | 34,493 | 495,924 | 14.38 | 8415 | - | 1 | 5-class clf. |
| TwitchEN | 7,126 | 77,774 | 10.91 | 128 | - | 1 | Binary clf. |
| CoauthorCS | 18,333 | 163,788 | 8.93 | 6805 | - | 1 | 15-class clf. |
| DBLP | 17,716 | 105,734 | 5.97 | 1639 | - | 1 | 4-class clf. |
| PubMed | 19,717 | 88,648 | 4.50 | 500 | - | 1 | 3-class clf. |
| Cora | 2,708 | 10,556 | 3.90 | 1433 | - | 1 | 7-class clf. |
| CiteSeer | 3,327 | 9,104 | 2.74 | 3703 | - | 1 | 6-class clf. |
| ogbn-arxiv | 169,343 | 1,166,243 | 6.89 | 128 | - | 1 | 40-class clf. |
| ogbn-products | 2,449,029 | 123,718,280 | 50.52 | 100 | - | 1 | 47-class clf. |
| ogbn-mag | 1,939,743 | 21,111,007 | 18.61 | 128 | 4 | 1 | 349-class clf. |
| ogbn-proteins | 132,534 | 79,122,504 | 597.00 | - | 8 | 112 | Multi-task |

use 4 message-passing layers, and thus the only difference in configuration lies in the learning rate.

We use residual connections between the GNN layers, 4 heads in the attention models, and the Parametric ReLU (PReLU) (He et al., 2015) as the nonlinear activation function. We do not use batch normalization (Ioffe & Szegedy, 2015), nor dropout (Srivastava et al., 2014). We use the Adam optimizer (Kingma & Ba, 2015) with $\beta = (0.9, 0.999)$, and an exponential learning-rate scheduler with $\gamma = 0.998$. We train all the models for 2500 epochs. Importantly, we do not use weight decay, since this will bias the solution towards $\lambda_1 = 0$ and $\lambda_2 = 1$.

We use the Pytorch Geometric (Fey & Lenssen, 2019) implementation of L-CAT for all experiments, switching between models by properly by setting $\lambda_1$ and $\lambda_2$. We parametrize $\lambda_1$ and $\lambda_2$ as free-parameters in log-space that pass through a sigmoid function—i.e., $\texttt{sigmoid}(10^x)$—such that they are constrained to the unit interval, and they are learned quickly.

### D.2 ADDITIONAL RESULTS

Table 6 shows the results presented in the main paper (with the addition of a dense feed-forward network), while Table 7 presents the results for the remaining datasets, with smaller average degree.

If we focus on Table 7, we observe that all models perform equally well, yet in a few cases CAT and L-CAT are significantly better than the baselines—e.g., L-CATv2 in *CoauthorCS*, or L-CAT in *Cora*. Following a similar discussion as the one presented in the main paper, these results indicates that L-CAT achieves similar or better performance than baseline models and thus, should be the preferred architecture.

**Competitive performance without the graph.** We also include in Tables 6 and 7 the performance of a feed-forward network, referred to as Dense (first row). Note that the only data available to this model are the node features, and thus no graph information is provided. Therefore, we should expect a significant drop in performance, which indeed happens for some datasets such as *Amazon Computers* ($\approx 7\%$ drop), *FacebookPagePage* ($\approx 20\%$ drop), *DBLP* ($\approx 9\%$ drop) and *Cora* ($\approx 14\%$ drop). Still, we found that for other commonly used datasets the performance is similar, e.g., *Coauthor Physics* and *PubMed*; or *it is even better CoauthorCS*. These results manifest the importance of a proper benchmarking, and of carefully considering the datasets used to evaluate GNN models.

Table 6: Test accuracy (%) of the considered convolution and attention models for different datasets (sorted by their average node degree), and averaged over ten runs. Bold numbers are statistically different to their baseline model ($\alpha = 0.05$). Best average performance is underlined.

| Dataset | Amazon Computers | Amazon Photo | GitHub | Facebook PagePage | Coauthor Physics | TwitchEN |
|---|---|---|---|---|---|---|
| Avg. Deg. | 35.76 | 31.13 | 15.33 | 15.22 | 14.38 | 10.91 |
| Dense | $83.73 \pm 0.34$ | $91.74 \pm 0.46$ | $81.21 \pm 0.30$ | $75.89 \pm 0.66$ | $95.41 \pm 0.14$ | $56.26 \pm 1.74$ |
| GCN | $\underline{90.59 \pm 0.36}$ | $\underline{95.13 \pm 0.57}$ | $84.13 \pm 0.44$ | $94.76 \pm 0.19$ | $96.36 \pm 0.10$ | $57.83 \pm 1.13$ |
| GAT | $89.59 \pm 0.61$ | $94.02 \pm 0.66$ | $83.31 \pm 0.18$ | $94.16 \pm 0.48$ | $96.36 \pm 0.10$ | $57.59 \pm 1.20$ |
| CAT | $\mathbf{90.58 \pm 0.40}$ | $\mathbf{94.77 \pm 0.47}$ | $\mathbf{84.11 \pm 0.66}$ | $\mathbf{94.71 \pm 0.30}$ | $\underline{96.40 \pm 0.10}$ | $\underline{58.09 \pm 1.61}$ |
| L-CAT | $\mathbf{90.34 \pm 0.47}$ | $\mathbf{94.93 \pm 0.37}$ | $84.05 \pm 0.70$ | $\underline{\mathbf{94.81 \pm 0.25}}$ | $96.35 \pm 0.10$ | $57.88 \pm 2.07$ |
| GATv2 | $89.49 \pm 0.53$ | $93.47 \pm 0.62$ | $82.92 \pm 0.45$ | $93.44 \pm 0.30$ | $96.24 \pm 0.19$ | $57.70 \pm 1.17$ |
| CATv2 | $\mathbf{90.44 \pm 0.46}$ | $\mathbf{94.81 \pm 0.55}$ | $\mathbf{84.10 \pm 0.88}$ | $\mathbf{94.27 \pm 0.31}$ | $96.34 \pm 0.12$ | $57.99 \pm 2.02$ |
| L-CATv2 | $\mathbf{90.33 \pm 0.44}$ | $\mathbf{94.79 \pm 0.61}$ | $\underline{84.31 \pm 0.59}$ | $\mathbf{94.44 \pm 0.39}$ | $96.29 \pm 0.13$ | $57.89 \pm 1.53$ |

Table 7: Test accuracy (%) of the considered convolution and attention models for different datasets (sorted by their average node degree), and averaged over ten runs. Bold numbers are statistically different to their baseline model ($\alpha = 0.05$). Best average performance is underlined.

| Dataset | CoauthorCS | DBLP | PubMed | Cora | CiteSeer |
|---|---|---|---|---|---|
| Avg. Deg. | 8.93 | 5.97 | 4.5 | 3.9 | 2.74 |
| Dense | $\underline{94.88 \pm 0.21}$ | $75.46 \pm 0.27$ | $88.13 \pm 0.33$ | $72.75 \pm 1.72$ | $73.02 \pm 1.01$ |
| GCN | $93.85 \pm 0.23$ | $84.18 \pm 0.40$ | $88.50 \pm 0.18$ | $\underline{86.68 \pm 0.78}$ | $\underline{75.76 \pm 1.09}$ |
| GAT | $93.80 \pm 0.38$ | $84.15 \pm 0.39$ | $88.62 \pm 0.18$ | $85.95 \pm 0.95$ | $75.40 \pm 1.43$ |
| CAT | $93.70 \pm 0.31$ | $84.10 \pm 0.29$ | $\underline{88.58 \pm 0.25}$ | $85.85 \pm 0.79$ | $75.64 \pm 0.91$ |
| L-CAT | $93.65 \pm 0.23$ | $84.13 \pm 0.26$ | $88.45 \pm 0.32$ | $\mathbf{86.66 \pm 0.87}$ | $75.04 \pm 1.12$ |
| GATv2 | $93.19 \pm 0.64$ | $84.33 \pm 0.18$ | $88.52 \pm 0.27$ | $85.65 \pm 1.01$ | $75.14 \pm 1.20$ |
| CATv2 | $93.51 \pm 0.34$ | $\mathbf{84.15 \pm 0.41}$ | $88.54 \pm 0.29$ | $85.50 \pm 0.94$ | $74.68 \pm 1.30$ |
| L-CATv2 | $\mathbf{93.65 \pm 0.20}$ | $84.31 \pm 0.31$ | $88.48 \pm 0.24$ | $85.75 \pm 0.72$ | $75.04 \pm 1.30$ |

# E  OPEN GRAPH BENCHMARK EXPERIMENTS

## E.1  EXPERIMENTAL DETAILS

**Computational resources.** For this set of experiments, we had at our disposal a set of 16 Tesla V100-SXM GPUs with 160 CPU cores, shared among the rest of the department.

**Statistical significance.** For each CAT and L-CAT model, we highlight significant improvements according to a two-sided paired t-test ($\alpha = 0.05$), with respect to its corresponding baseline model. For example, for L-CATv2 with 8 heads we perform the test with respect to GATv2 with 8 heads.

**General experimental setup.** As mentioned in §6.3, we repeat all experiments with OGB datasets 5 times. In all cases, we choose the model parameters with the best validation performance during training. Moreover, when we show the results without specifying the number of heads, we take the model with the best validation error among the two models with 1 and 8 heads.

We use the same implementation of L-CAT for all experiments, switching between models by properly setting $\lambda_1$ and $\lambda_2$. Experiments on *arxiv*, *mag*, *products* use a version of L-CAT implemented in Pytorch Geometric (Fey & Lenssen, 2019). Experiments on *proteins* use a version of L-CAT implemented in DGL (Wang et al., 2019a). We parametrize $\lambda_1$ and $\lambda_2$ as free-parameters in log-space that pass through a sigmoid function—i.e., $\texttt{sigmoid}(10^x)$—such that they are constrained to the unit interval, and they are learned quickly.

**ArXiv.** As described in §6.3, we use the example code from the OGB framework (Hu et al., 2020a). The network is composed of 3 GNN layers with a hidden size of 128. We use batch normalization (Ioffe & Szegedy, 2015) and a dropout (Srivastava et al., 2014) of 0.5 between the GNN layers, and Adam (Kingma & Ba, 2015) with a learning rate of 0.01. We use the ReLU as activation function. For the initial experiments, we train for 1500 epochs, while we train for 500 epochs for the noise experiments in §6.3.1. This is justified given the convergence plots in Fig. 2.

**MAG.** We adapted the official code from (Brody et al., 2022). The network is composed of 2 layers with 128 hidden channels. This time, we use layer normalization (Ba et al., 2016) and a dropout of 0.5 between the layers. Again, we use ReLU as the activation function, and add residual connections to the network. As with *arxiv*, we use Adam (Kingma & Ba, 2015) with learning rate 0.01. We set a batch size of 20000 and train for 100 epochs.

**Products.** We use the same setup as (Brody et al., 2022), with a network of 3 GNN layers and 128 hidden dimensions. We apply residual connections once again, with a dropout (Srivastava et al., 2014) of 0.5 between layers. This time, we use ELU as the activation function. The batch size is set to 256. Adam (Kingma & Ba, 2015) is again the optimizer in use, this time with a learning rate of 0.001. We train for 100 epochs, although we apply early stopping whenever the validation accuracy stops increasing for more than 10 epochs. Note the training split of this dataset only contains 8 % of the data.

**Proteins.** We follow once more the setup of (Brody et al., 2022). The network we use has 6 GNN layers of hidden size 64. Dropout (Srivastava et al., 2014) is set to 0.25 between layers, with an input dropout of 0.1. At the beginning of the network, we place a linear layer followed by a ReLU activation to encode the nodes, and a linear layer at the end of the network to predict the class. Moreover, we use batch normalization (Ioffe & Szegedy, 2015) between layers and ReLU as the activation function. We train the model for 1200 epochs at most, with early stopping after not improving for 10 epochs.

### E.2 Additional results

We show in Tables 8 to 10 the results of the main paper for the *arxiv*, *mag*, *products* datasets, respectively, without selecting the best configuration for each type of model. That is, we show the results for both number of heads. Note that we already show the full table of results for the *protein* datasets in the main paper (Table 3). All the trends discussed in the main paper hold.

Table 8: Test accuracy on the *arxiv* dataset for attention models using 1 head and 8 heads.

|  | GCN | GAT | CAT | L-CAT | GATv2 | CATv2 | L-CATv2 |
|---|---|---|---|---|---|---|---|
| 1h | 71.58 ± 0.19 | 71.58 ± 0.15 | **72.04 ± 0.20** | **72.00 ± 0.11** | 71.70 ± 0.14 | **72.02 ± 0.08** | 71.96 ± 0.21 |
| 8h | – | 71.63 ± 0.11 | **72.14 ± 0.20** | **71.98 ± 0.08** | 71.72 ± 0.24 | 71.76 ± 0.14 | 71.91 ± 0.16 |

Table 9: Test accuracy on the *mag* dataset for attention models using 1 head and 8 heads.

|  | GCN | GAT | CAT | L-CAT | GATv2 | CATv2 | L-CATv2 |
|---|---|---|---|---|---|---|---|
| 1h | 32.77 ± 0.36 | 32.35 ± 0.24 | 31.98 ± 0.46 | 32.47 ± 0.38 | 32.76 ± 0.18 | **32.43 ± 0.22** | 32.68 ± 0.50 |
| 8h | – | 32.15 ± 0.31 | **31.58 ± 0.22** | 32.49 ± 0.21 | 32.85 ± 0.21 | **32.34 ± 0.18** | **32.38 ± 0.28** |

Table 10: Test accuracy on the *products* dataset for attention models using 1 head and 8 heads.

|  | GCN | GAT | CAT | L-CAT | GATv2 | CATv2 | L-CATv2 |
|---|---|---|---|---|---|---|---|
| 1h | 74.12 ± 1.20 | 78.53 ± 0.91 | **77.38 ± 0.36** | 77.19 ± 1.11 | 73.81 ± 0.39 | 74.81 ± 1.12 | **76.37 ± 0.92** |
| 8h | – | 78.23 ± 0.25 | **76.63 ± 1.15** | **76.56 ± 0.45** | 76.40 ± 0.71 | 75.20 ± 0.92 | **74.70 ± 0.28** |

**Extrapolation ablation study.** Due to page constraints, these results were not added to the main paper. Here, we study two questions. First, how important are $\lambda_1$ and $\lambda_2$ in the formulation of L-CAT (Eq. 7)? For the sake of completeness, the second question we

attempt to answer here is whether we can obtain similar performance by just interpolating between GCN and GAT (fixing $\lambda_2 = 0$)? Note that we theoretically showed in §§4 and 6.1 that CAT fills up a gap between GCN and GAT, making it preferable in certain settings.

To this end, we repeat the experiments for network-initialization robustness in §6.3.2, since they showed to be the best ones to tell apart the performance across models. We include three additional models: GCN-GAT, which interpolates between GCN and GAT (or GATv2) by learning $\lambda_1$ and fixing $\lambda_2 = 0$; CAT-$\lambda_1$ which interpolates between GCN and CAT by learning $\lambda_1$ and fixing $\lambda_2 = 1$; and CAT-$\lambda_2$, which interpolates between GAT and CAT by learning $\lambda_2$ and fixing $\lambda_1 = 1$.

Results using GAT and shown in Table 11, and using GATv2 in Table 12. We can observe that GCN-GAT obtains results in between GCN and GAT for all settings, despite being able to interpolate between both layers in each of the six layers of the network. Regarding learning $\lambda_1$ and $\lambda_2$, we can observe that there is a clear difference between learning boths (L-CAT), and learning a single one. For both attention models, CAT-$\lambda_1$ obtains better results than CAT-$\lambda_2$ in all settings, but *uniform* with 8 heads. Still, the results of both variants are substantially worse than those of L-CAT in all cases, *demonstrating the importance of learning to interpolate between the three layer types.*

Table 11: Test accuracy on the *proteins* dataset for GCN (Kipf & Welling, 2017) and GAT (Velickovic et al., 2018) attention models using two network initializations, and two numbers of heads (1 and 8).

|  | GCN | GCN-GAT | GAT | CAT | L-CAT | CAT-$\lambda_1$ | CAT-$\lambda_2$ |
|---|---|---|---|---|---|---|---|
|  |  |  |  | *uniform* initialization |  |  |  |
| 1h | $61.08 \pm 2.86$ | $\mathbf{70.44 \pm 1.56}$ | $59.73 \pm 4.04$ | $\mathbf{74.19 \pm 0.72}$ | $\mathbf{77.77 \pm 1.44}$ | $71.97 \pm 3.78$ | $\mathbf{73.55 \pm 1.36}$ |
| 8h | $-$ | $\mathbf{68.51 \pm 0.91}$ | $72.23 \pm 3.20$ | $73.60 \pm 1.27$ | $\underline{\mathbf{78.85 \pm 1.76}}$ | $\mathbf{76.43 \pm 2.47}$ | $72.76 \pm 2.79$ |
|  |  |  |  | *normal* initialization |  |  |  |
| 1h | $\underline{80.10 \pm 0.61}$ | $66.51 \pm 3.23$ | $66.38 \pm 7.76$ | $73.26 \pm 1.84$ | $\mathbf{78.06 \pm 1.40}$ | $\mathbf{76.77 \pm 1.91}$ | $73.39 \pm 1.25$ |
| 8h | $-$ | $\mathbf{69.93 \pm 1.93}$ | $79.08 \pm 1.64$ | $\mathbf{74.67 \pm 1.29}$ | $\underline{79.63 \pm 0.79}$ | $78.86 \pm 1.07$ | $\mathbf{73.32 \pm 1.15}$ |

Table 12: Test accuracy on the *proteins* dataset for GCN (Kipf & Welling, 2017) and GATv2 (Brody et al., 2022) attention models using two network initializations, and two numbers of heads (1 and 8).

|  | GCN | GCN-GATv2 | GATv2 | CATv2 | L-CATv2 | CATv2-$\lambda_1$ | CATv2-$\lambda_2$ |
|---|---|---|---|---|---|---|---|
|  |  |  |  | *uniform* initialization |  |  |  |
| 1h | $61.08 \pm 2.86$ | $\mathbf{69.69 \pm 1.59}$ | $59.85 \pm 3.05$ | $\mathbf{64.32 \pm 2.61}$ | $\mathbf{79.08 \pm 1.06}$ | $63.24 \pm 1.55$ | $\mathbf{73.41 \pm 0.34}$ |
| 8h | $-$ | $\mathbf{69.94 \pm 1.62}$ | $75.21 \pm 1.80$ | $74.16 \pm 1.45$ | $\underline{78.77 \pm 1.09}$ | $\mathbf{77.61 \pm 1.32}$ | $73.96 \pm 1.27$ |
|  |  |  |  | *normal* initialization |  |  |  |
| 1h | $\underline{80.10 \pm 0.61}$ | $68.54 \pm 1.63$ | $69.13 \pm 9.48$ | $74.33 \pm 1.06$ | $\mathbf{79.07 \pm 1.09}$ | $78.41 \pm 0.93$ | $74.07 \pm 1.17$ |
| 8h | $-$ | $\mathbf{68.71 \pm 1.96}$ | $78.65 \pm 1.61$ | $\mathbf{73.40 \pm 0.62}$ | $\underline{79.30 \pm 0.55}$ | $78.76 \pm 1.41$ | $\mathbf{73.22 \pm 0.77}$ |

## F  EXTENDING L-CAT TO OTHER GNN MODELS

Due to their simplicity and popularity, in the manuscript we focus on the simplest form of GCNs, as described in §2. However, we consider important to remark that L-CAT can be effortless extended to a large range of existing GNN models. A more general formulation of a message-passing GNN layer than the one given in Eq. 1 is the following:

$$\tilde{\boldsymbol{h}}_i = f(\boldsymbol{h}'_i) \quad \text{where} \quad \boldsymbol{h}'_i \overset{\text{def}}{=} \bigoplus_{j \in N_i^*} \widehat{\gamma}_{ij} M\left(\boldsymbol{h}_i, \boldsymbol{h}_j; \theta_M\right) , \tag{13}$$

where $\widehat{\gamma}_{ij}$ is a scalar value, $\bigoplus$ refers to any permutation invariant operation—e.g., sum, mean, maximum, or minimum operations—and $M$ is the message operator, which can be parameterized, and produces a message based on the sender and receiver representations. This formulation comprises most GNN architectures present in the current literature. For example:

- GCN (Kipf & Welling, 2017): $\boldsymbol{h}'_i = \sum_{j \in N_i^*} \widehat{\gamma}_{ij} \boldsymbol{W}_v \boldsymbol{h}_j$ where $\widehat{\gamma}_{ij} = \frac{1}{|N_i^*|}$ as consider in the main paper, or $\widehat{\gamma}_{ij} = \frac{1}{\sqrt{d_j d_i}}$, where $d_i$ is the number of neighbors of node $i$ (including self-loops), if we consider the symmetric normalized adjacency matrix instead.
- GIN (Xu et al., 2019): $\boldsymbol{h}'_i = (1 + \varepsilon)\widehat{\gamma}_{ii}\boldsymbol{h}_i + \sum_{j \in N_i} \widehat{\gamma}_{ij}\boldsymbol{h}_j$ with $\widehat{\gamma}_{ij} = 1$.
- PNA (Corso et al., 2020): $\boldsymbol{h}'_i = \bigoplus_{j \in N_i^*} \widehat{\gamma}_{ij} M\left(\boldsymbol{h}_i, \boldsymbol{h}_j; \theta_M\right)$ where $\widehat{\gamma}_{ij} = 1$, $M$ is an multi-layer perceptron, and $\bigoplus$ is a set of permutation invariant operations, e.g., $\bigoplus = [\mu, \sigma, \max, \min]$.
- GCNII (Chen et al., 2020): $\boldsymbol{h}'_i = \left(\alpha\boldsymbol{h}_i^0 + (1 - \alpha)\sum_{j \in N_i^*} \widehat{\gamma}_{ij}\boldsymbol{h}_j\right)((1 - \beta)\boldsymbol{I} + \beta\boldsymbol{W}_v)$ where, just as in the GCN case, $\widehat{\gamma}_{ij} = \frac{1}{\sqrt{d_j d_i}}$, and $0 \leq \alpha, \beta \leq 1$.

In all the models above, the values $\widehat{\gamma}_{ij}$ are taken from the adjacency matrix $A$ (whose entries are 1 if there exists an edge between nodes $i$ and $j$, and 0 otherwise), or a matrix derived from it, e.g., the symmetric normalized adjacency matrix.

Note that the attention coefficients $\gamma_{ij}$ defined in Eq. 3 can be understood as an attention-equivalent of the adjacency matrix. Indeed, by defining $A^{att}$ as a matrix whose entries are $|N_i^*|\gamma_{ij}$, one can obtain a row-stochastic matrix that can substitute the adjacency matrix of any GNN model. This technique to generalize attention models to GNN variants more complex than a GCN has been successfully applied in prior literature (Wang et al., 2021b).

With this new interpretation of attention-models, the interpolation performed by L-CAT (see Eq. 7) can similarly be re-interpreted. Indeed, L-CAT learns to interpolate between the adjacency matrix $A$, and an attention-based adjacency matrix $A^{att}$, which can be produced by either GAT (Eq. 3) or CAT (Eq. 6), depending on the value of $\lambda_2$.

### F.1  PNA EXPERIMENTS

In Tables 13 and 14, we show the results—for the datasets described in App. C—using L-CAT and CAT in conjunction with the PNA model (Corso et al., 2020). First, we note that standard PNA works quite well in most cases. Second, if we focus on Table 13, we observe that the standard attention models (i.e., PNAGAT and PNAGATv2) perform significantly worse than the other approaches, in particular on datasets with large average degree, e.g., on *Amazon Computers*. Finally, we observe that the L-CAT models (i.e., L-PNACAT and L-PNACATv2) drastically improve the performance of their attention counterparts and achieve similar performance as the PNA model, with lower performance on the *GitHub* and *Facebook* datasets and higher performance on *Cora* and *CiteSeer*.

To keep the same number of parameters, we reuse the PNA weights to compute the attention scores ($\boldsymbol{W}_q = \boldsymbol{W}_k = \boldsymbol{W}_v$). However, this could be detrimental, as the role of $\boldsymbol{W}_v$ is completely different from that of $\boldsymbol{W}_q$ and $\boldsymbol{W}_v$. Tables 15 and 16 show the same experiments as before, but using different parameters to compute the keys and queries (i.e., $\boldsymbol{W}_q = \boldsymbol{W}_k \neq \boldsymbol{W}_v$). We

observe that the increase of parameters generally helps both CAT and L-CAT models, now outperform the base PNA model in some settings.

Table 13: Test accuracy (%) of the PNA (Corso et al., 2020) models for different datasets (sorted by average node degree), averaged over ten runs. Bold numbers are statistically different to their baseline model ($\alpha = 0.05$). Best average performance is underlined.

| Dataset | Amazon Computers | Amazon Photo | GitHub | Facebook PagePage | Coauthor Physics | TwitchEN |
|---|---|---|---|---|---|---|
| Avg. Deg. | 35.76 | 31.13 | 15.33 | 15.22 | 14.38 | 10.91 |
| PNA | $\underline{86.51 \pm 1.22}$ | $93.23 \pm 0.65$ | $82.33 \pm 0.51$ | $94.28 \pm 0.34$ | $96.09 \pm 0.14$ | $\underline{59.25 \pm 1.19}$ |
| PNAGAT | $57.59 \pm 10.19$ | $74.78 \pm 8.74$ | $72.77 \pm 2.06$ | $71.49 \pm 11.23$ | $96.05 \pm 0.25$ | $54.22 \pm 3.02$ |
| PNACAT | $\mathbf{81.48 \pm 3.81}$ | $\mathbf{91.73 \pm 1.24}$ | $75.55 \pm 3.33$ | $\mathbf{93.10 \pm 0.41}$ | $96.16 \pm 0.15$ | $\mathbf{59.11 \pm 1.94}$ |
| L-PNACAT | $86.45 \pm 1.42$ | $92.76 \pm 0.74$ | $\mathbf{78.74 \pm 2.91}$ | $\mathbf{93.59 \pm 0.39}$ | $\underline{96.24 \pm 0.13}$ | $59.12 \pm 2.74$ |
| PNAGATv2 | $36.93 \pm 4.07$ | $60.13 \pm 4.81$ | $73.93 \pm 1.89$ | $58.91 \pm 3.42$ | $95.61 \pm 0.29$ | $54.45 \pm 1.60$ |
| PNACATv2 | $\mathbf{79.08 \pm 2.62}$ | $\mathbf{88.61 \pm 3.24}$ | $75.11 \pm 2.79$ | $\mathbf{92.77 \pm 0.50}$ | $96.06 \pm 0.18$ | $\mathbf{56.72 \pm 2.43}$ |
| L-PNACATv2 | $\mathbf{85.10 \pm 1.70}$ | $\mathbf{92.19 \pm 0.55}$ | $\mathbf{79.79 \pm 1.40}$ | $\mathbf{93.54 \pm 0.36}$ | $96.03 \pm 0.19$ | $\mathbf{58.19 \pm 1.53}$ |

Table 14: Test accuracy (%) of the PNA (Corso et al., 2020) models for different datasets (sorted by average node degree), averaged over ten runs. Bold numbers are statistically different to their baseline model ($\alpha = 0.05$). Best average performance is underlined.

| Dataset | CoauthorCS | DBLP | PubMed | Cora | CiteSeer |
|---|---|---|---|---|---|
| Avg. Deg. | 8.93 | 5.97 | 4.5 | 3.9 | 2.74 |
| PNA | $\underline{93.30 \pm 0.31}$ | $83.37 \pm 0.32$ | $88.37 \pm 0.73$ | $84.94 \pm 1.19$ | $73.92 \pm 0.97$ |
| PNAGAT | $92.46 \pm 0.95$ | $83.42 \pm 0.39$ | $88.40 \pm 0.33$ | $84.67 \pm 0.69$ | $74.64 \pm 0.82$ |
| PNACAT | $92.90 \pm 0.24$ | $83.35 \pm 0.40$ | $\underline{88.24 \pm 0.30}$ | $\mathbf{85.58 \pm 1.00}$ | $74.94 \pm 1.68$ |
| L-PNACAT | $93.11 \pm 0.24$ | $83.21 \pm 0.55$ | $88.22 \pm 0.40$ | $\underline{\mathbf{85.77}} \pm 1.01$ | $\underline{75.08 \pm 1.05}$ |
| PNAGATv2 | $90.14 \pm 0.82$ | $83.37 \pm 0.34$ | $88.14 \pm 0.45$ | $85.04 \pm 0.86$ | $74.50 \pm 1.18$ |
| PNACATv2 | $\mathbf{92.78 \pm 0.27}$ | $83.22 \pm 0.38$ | $88.28 \pm 0.30$ | $85.41 \pm 0.98$ | $74.42 \pm 1.11$ |
| L-PNACATv2 | $\mathbf{93.02 \pm 0.37}$ | $\underline{83.54 \pm 0.65}$ | $88.23 \pm 0.58$ | $85.48 \pm 0.98$ | $74.76 \pm 1.57$ |

Table 15: Test accuracy (%) of the PNA (Corso et al., 2020) extended models with $\boldsymbol{W}_q = \boldsymbol{W}_k \neq \boldsymbol{W}_v$ for different datasets, averaged over ten runs. Bold numbers are statistically different to their baseline model ($\alpha = 0.05$). Best average performance is underlined.

| Dataset | Amazon Computers | Amazon Photo | GitHub | Facebook PagePage | Coauthor Physics | TwitchEN |
|---|---|---|---|---|---|---|
| Avg. Deg. | 35.76 | 31.13 | 15.33 | 15.22 | 14.38 | 10.91 |
| PNA | $\underline{86.51 \pm 1.22}$ | $93.23 \pm 0.65$ | $\underline{82.33 \pm 0.51}$ | $94.28 \pm 0.34$ | $96.09 \pm 0.14$ | $\underline{59.25 \pm 1.19}$ |
| PNAGAT | $48.65 \pm 19.25$ | $68.01 \pm 20.32$ | $72.97 \pm 1.07$ | $70.17 \pm 12.02$ | $96.02 \pm 0.34$ | $53.27 \pm 2.54$ |
| PNACAT | $\mathbf{83.45 \pm 2.60}$ | $\mathbf{91.62 \pm 1.30}$ | $75.35 \pm 2.71$ | $\mathbf{93.31 \pm 0.55}$ | $96.22 \pm 0.13$ | $\mathbf{59.23 \pm 2.25}$ |
| L-PNACAT | $87.18 \pm 1.22$ | $92.79 \pm 0.63$ | $\mathbf{79.64 \pm 2.54}$ | $\mathbf{93.78 \pm 0.39}$ | $\mathbf{96.31 \pm 0.17}$ | $59.09 \pm 2.50$ |
| PNAGATv2 | $39.49 \pm 4.09$ | $62.19 \pm 11.30$ | $73.97 \pm 1.67$ | $63.00 \pm 4.95$ | $95.83 \pm 0.36$ | $55.21 \pm 1.05$ |
| PNACATv2 | $\mathbf{81.20 \pm 3.63}$ | $\mathbf{91.32 \pm 0.80}$ | $74.57 \pm 2.18$ | $\mathbf{92.98 \pm 0.36}$ | $\mathbf{96.14 \pm 0.16}$ | $56.21 \pm 2.01$ |
| L-PNACATv2 | $\mathbf{86.22 \pm 0.83}$ | $\mathbf{92.98 \pm 0.89}$ | $\mathbf{79.78 \pm 2.48}$ | $\mathbf{93.44 \pm 0.37}$ | $96.13 \pm 0.12$ | $\mathbf{60.26 \pm 1.25}$ |

Table 16: Test accuracy (%) of the PNA (Corso et al., 2020) extended models with $W_q = W_k \neq W_v$ for different datasets, averaged over ten runs. Bold numbers are statistically different to their baseline model ($\alpha = 0.05$). Best average performance is underlined.

| Dataset | CoauthorCS | DBLP | PubMed | Cora | CiteSeer |
|---|---|---|---|---|---|
| Avg. Deg. | 8.93 | 5.97 | 4.5 | 3.9 | 2.74 |
| PNA | 93.30 ± 0.31 | 83.37 ± 0.32 | 88.37 ± 0.73 | 84.94 ± 1.19 | 73.92 ± 0.97 |
| PNAGAT | 92.50 ± 0.46 | 83.22 ± 0.45 | 88.43 ± 0.29 | 84.89 ± 1.15 | 75.76 ± 1.29 |
| PNAGAT | **92.97 ± 0.47** | 83.28 ± 0.59 | 88.27 ± 0.43 | 85.09 ± 0.70 | 75.44 ± 1.51 |
| PNAGAT | **93.17 ± 0.30** | 83.50 ± 0.29 | 88.54 ± 0.45 | 85.63 ± 0.92 | 75.22 ± 1.12 |
| PNAGATv2 | 90.00 ± 1.01 | 83.40 ± 0.48 | 88.14 ± 0.31 | 85.16 ± 0.91 | 76.14 ± 1.33 |
| PNAGATv2 | **92.74 ± 0.20** | **83.05 ± 0.49** | **88.38 ± 0.31** | 85.21 ± 0.83 | 75.80 ± 1.26 |
| PNAGATv2 | **93.02 ± 0.30** | 83.24 ± 0.44 | 88.28 ± 0.35 | 85.04 ± 0.94 | 75.80 ± 1.19 |

Table 17: Test accuracy (%) of the GCNII (Chen et al., 2020) models for different datasets, averaged over ten runs. Bold numbers are statistically different to their baseline model ($\alpha = 0.05$). Best average performance is underlined.

| Dataset | Amazon Computers | Amazon Photo | GitHub | Facebook PagePage | Coauthor Physics | TwitchEN |
|---|---|---|---|---|---|---|
| Avg. Deg. | 35.76 | 31.13 | 15.33 | 15.22 | 14.38 | 10.91 |
| GCNII | 90.82 ± 0.20 | 95.51 ± 0.48 | 84.11 ± 0.76 | 94.03 ± 0.30 | 96.58 ± 0.11 | 60.94 ± 1.66 |
| GCNIIGAT | 89.04 ± 0.87 | 94.74 ± 0.57 | 82.34 ± 0.64 | 91.18 ± 0.82 | 96.69 ± 0.13 | 57.76 ± 1.76 |
| GCNIICAT | **89.83 ± 0.42** | **95.31 ± 0.25** | **83.15 ± 0.51** | **93.25 ± 0.37** | 96.69 ± 0.09 | **60.51 ± 1.12** |
| L-GCNIICAT | **90.03 ± 0.42** | 95.23 ± 0.39 | **83.50 ± 0.57** | **93.71 ± 0.33** | **96.87 ± 0.14** | **61.14 ± 1.64** |
| GCNIIGATv2 | 84.26 ± 2.80 | 89.23 ± 5.30 | 81.23 ± 0.45 | 83.82 ± 1.24 | 96.14 ± 0.28 | 56.25 ± 1.56 |
| GCNIICATv2 | **89.59 ± 0.45** | **95.03 ± 0.55** | **82.45 ± 0.30** | **92.55 ± 0.52** | **96.50 ± 0.09** | **59.04 ± 1.49** |
| L-GCNIICATv2 | **89.81 ± 0.48** | **95.24 ± 0.35** | **83.05 ± 0.49** | **93.68 ± 0.35** | **96.75 ± 0.11** | **61.10 ± 1.11** |

## F.2 GCNII EXPERIMENTS

Similarly, we have run the experiments from §6.2, this time combining GCNII (Chen et al., 2020) with GAT, CAT, and L-CAT as explained above. Results are shown in Tables 17 and 18, in which we can observe that the baseline model obtains the best results so far in the manuscript (in comparison with both GCN and PNA). And just as before, we observe again that CAT and L-CAT always improve with respect to their base models, staying on par with the baseline GNCII model and, sometimes, even outperforming the baseline model on average (e.g., *Coauthor Physics*, *TwitchEN*). As with the experiments for PNA, Tables 19 and 20 shows the results when the attention matrices are different from the value matrix ($W_q = W_k \neq W_v$). We can similarly observe that most of the results are improved with the additional parameters, beating the baseline model in different datasets.

Table 18: Test accuracy (%) of the GCNII (Chen et al., 2020) models for different datasets, averaged over ten runs. Bold numbers are statistically different to their baseline model ($\alpha = 0.05$). Best average performance is underlined.

| Dataset | CoauthorCS | DBLP | PubMed | Cora | CiteSeer |
|---|---|---|---|---|---|
| Avg. Deg. | 8.93 | 5.97 | 4.5 | 3.9 | 2.74 |
| GCNII | 95.36 ± 0.18 | 83.86 ± 0.14 | 89.05 ± 0.28 | 86.49 ± 0.79 | 76.46 ± 0.71 |
| GCNIIGAT | 95.32 ± 0.27 | 83.45 ± 0.60 | 88.24 ± 0.34 | 85.72 ± 1.05 | 75.78 ± 0.77 |
| GCNIICAT | 95.12 ± 0.25 | 83.86 ± 0.23 | 88.65 ± 0.40 | 85.51 ± 0.95 | **76.60 ± 0.50** |
| L-GCNIICAT | 95.30 ± 0.29 | 83.76 ± 0.26 | **88.72 ± 0.35** | 85.48 ± 0.96 | **76.76 ± 0.51** |
| GCNIIGATv2 | 93.37 ± 0.73 | 83.70 ± 0.42 | 88.49 ± 0.34 | 85.82 ± 1.35 | 75.98 ± 0.71 |
| GCNIICATv2 | **95.01 ± 0.32** | 83.93 ± 0.36 | 88.60 ± 0.27 | 85.72 ± 1.03 | 76.62 ± 0.63 |
| L-GCNIICATv2 | **95.29 ± 0.23** | 83.93 ± 0.41 | **89.12 ± 0.35** | 85.73 ± 0.88 | 76.22 ± 0.98 |

Table 19: Test accuracy (%) of the GCNII (Chen et al., 2020) extended models with $\boldsymbol{W}_q = \boldsymbol{W}_k \neq \boldsymbol{W}_v$ for different datasets, averaged over ten runs. Bold numbers are statistically different to their baseline model ($\alpha = 0.05$). Best average performance is underlined.

| Dataset | Amazon Computers | Amazon Photo | GitHub | Facebook PagePage | Coauthor Physics | TwitchEN |
|---|---|---|---|---|---|---|
| Avg. Deg. | 35.76 | 31.13 | 15.33 | 15.22 | 14.38 | 10.91 |
| GCNII | $90.82 \pm 0.20$ | $95.51 \pm 0.48$ | $84.11 \pm 0.76$ | $94.03 \pm 0.30$ | $96.58 \pm 0.11$ | $60.94 \pm 1.66$ |
| GCNIIGAT | $89.24 \pm 0.59$ | $94.66 \pm 0.59$ | $82.45 \pm 0.65$ | $90.90 \pm 0.71$ | $96.90 \pm 0.16$ | $58.12 \pm 2.02$ |
| GCNIICAT | $\mathbf{89.94 \pm 0.40}$ | $95.03 \pm 0.37$ | $\mathbf{83.12 \pm 0.37}$ | $\mathbf{93.39 \pm 0.31}$ | $\underline{\mathbf{96.59 \pm 0.07}}$ | $59.60 \pm 0.76$ |
| L-GCNIICAT | $\mathbf{90.35 \pm 0.46}$ | $\underline{\mathbf{95.53 \pm 0.35}}$ | $83.48 \pm 0.47$ | $93.63 \pm 0.39$ | $96.80 \pm 0.09$ | $60.77 \pm 2.15$ |
| GCNIIGATv2 | $85.70 \pm 2.58$ | $91.24 \pm 2.43$ | $81.43 \pm 0.39$ | $84.59 \pm 0.79$ | $96.55 \pm 0.20$ | $55.23 \pm 2.15$ |
| GCNIICATv2 | $\mathbf{89.56 \pm 0.67}$ | $\mathbf{95.46 \pm 0.54}$ | $\mathbf{82.50 \pm 0.47}$ | $\mathbf{93.04 \pm 0.40}$ | $96.55 \pm 0.10$ | $\mathbf{59.57 \pm 1.45}$ |
| L-GCNIICATv2 | $\mathbf{90.24 \pm 0.32}$ | $\underline{\mathbf{95.53 \pm 0.28}}$ | $\mathbf{83.34 \pm 0.45}$ | $\mathbf{93.67 \pm 0.47}$ | $96.73 \pm 0.14$ | $\mathbf{60.65 \pm 1.13}$ |

Table 20: Test accuracy (%) of the GCNII (Chen et al., 2020) extended models with $\boldsymbol{W}_q = \boldsymbol{W}_k \neq \boldsymbol{W}_v$ for different datasets, averaged over ten runs. Bold numbers are statistically different to their baseline model ($\alpha = 0.05$). Best average performance is underlined.

| Dataset | CoauthorCS | DBLP | PubMed | Cora | CiteSeer |
|---|---|---|---|---|---|
| Avg. Deg. | 8.93 | 5.97 | 4.5 | 3.9 | 2.74 |
| GCNII | $95.36 \pm 0.18$ | $83.86 \pm 0.14$ | $\underline{89.05 \pm 0.28}$ | $\underline{86.49 \pm 0.79}$ | $76.46 \pm 0.71$ |
| GCNIIGAT | $95.36 \pm 0.20$ | $83.60 \pm 0.32$ | $88.33 \pm 0.35$ | $85.26 \pm 1.19$ | $76.30 \pm 0.78$ |
| GCNIICAT | $\mathbf{95.20 \pm 0.12}$ | $\underline{\mathbf{83.89 \pm 0.29}}$ | $88.45 \pm 0.29$ | $86.44 \pm 1.22$ | $76.70 \pm 0.60$ |
| L-GCNIICAT | $\underline{95.47 \pm 0.16}$ | $83.70 \pm 0.40$ | $88.08 \pm 0.47$ | $86.02 \pm 1.43$ | $76.54 \pm 0.59$ |
| GCNIIGATv2 | $93.97 \pm 0.57$ | $83.67 \pm 0.24$ | $88.24 \pm 0.16$ | $85.28 \pm 1.11$ | $76.58 \pm 0.64$ |
| GCNIICATv2 | $\mathbf{95.05 \pm 0.33}$ | $83.78 \pm 0.35$ | $88.35 \pm 0.34$ | $\mathbf{86.49 \pm 0.90}$ | $\mathbf{75.28 \pm 0.84}$ |
| L-GCNIICATv2 | $\underline{\mathbf{95.45 \pm 0.18}}$ | $83.86 \pm 0.24$ | $88.06 \pm 0.32$ | $\underline{86.49 \pm 1.31}$ | $76.80 \pm 0.42$ |

