# OpenReview forum: "Learnable Graph Convolutional Attention Networks"
_ICLR.cc/2023/Conference — ICLR 2023 poster_

### Official Review · Reviewer_CSVs · 2022-10-25

**Confidence:** 4
**Clarity, Quality, Novelty And Reproducibility:** Please see the above reviews.
**Correctness:** 3
**Technical Novelty And Significance:** 2
**Empirical Novelty And Significance:** Not applicable
**Recommendation:** 6

**Strength And Weaknesses:**

Strength

1, The motivation that different data may need different aggregation makes sense.

2, The analysis of the evaluation of $\lambda_1$ and $\lambda_2$ is interesting.

Weaknesses

1, The comparison is insufficient. Only two methods are compared on most of the datasets: GCN and GAT. The weak comparison makes it hard to support the effectiveness of the proposed method.

2, The novelty is limited. The core idea of the proposed method is to integrate several different aggregation strategies. There are already several research works along this line like the sampling method proposed in [1].

3, In terms of technical detail, it seems that the proposed method is designed only for native GCN and GAT. It is hard to generalize the proposed method to other kinds of aggregations. It is more like another new aggregation scheme, which limits the contribution.

Other issues:

The highlights of Table.2 are improper.  E.g. the GCN performs the best on proteins while the L-CATv2 is highlighted.

[1] Meta-Aggregator: Learning to Aggregate for 1-bit Graph Neural Networks, ICCV 2021

**Summary Of The Paper:**

This manuscript focuses on the problem of learning better aggregation attention. The motivation comes from the observation that there is no clear winner of the existing three aggregation methods, including native GCN and GAT. The authors propose a new CAT aggregation method and merge it with GCN and GAT. The proposed method is evaluated on several standard datasets to show its effectiveness.

**Summary Of The Review:**

In short, the main concerns of this manuscript are two parts: the insufficient evaluation and the limited novetly.

---

> ### Author Response · Authors · 2022-11-09
> **Reply to reviewer CSVs**
>
> Foremost, many thanks for the thought-provoking review. Below, we address/clarify each of the individual items.
>
> - **Core idea and novelty.** From our point of view, (L-)CAT does not work at the aggregation level, but rather at the graph level, as attention can be interpreted as obtaining a (weighted) sub-graph of the original graph. In particular, L-CAT learns to interpolate between different *adjacency matrices* based on the training data, which to our eyes, is a novel contribution. While this is discussed at the beginning of Appendix F, we will work to make this point clearer in the main paper.
> - **Generalize L-CAT to other approaches.** Following the previous point, L-CAT can be applied to any GNN approach that uses an adjacency matrix in their formulation, such as the suggested GNA and ANA methods, PNA, or GCNII (see Appendix F). In the main text, we stick to the native GCN for ease of exposition, as well as to keep it consistent with the model assumed in the theoretical findings, but in Appendix F we show how to apply it to other models obtaining consistent empirical results with those on the main paper
> - **Experiment comparisons.** The main goal of our experiments is to show that CAT is a viable GNN layer, that L-CAT is a highly flexible and more robust approach than its fixed instantiations (i.e., GCN, GAT, and CAT), and that in doing so L-CAT does not give up on performance, which we believe we properly demonstrate (see, e.g., Table 3 for a clear example showing that L-CAT obtains *for all configurations* comparable results to the best model, and Figure 3 to see the flexibility that allows L-CAT achieve these results). We have updated the first paragraph of section 6 to make this point clear. The adaptability of L-CAT to other GNN/GAT models is also important and, for this reason, we consider two GAT methods in the main paper, and two GNN baselines (GCN in the main paper, and PNA in Appendix F.1). Additionally, we have expanded the results in appendix F.1 to include GCNII as a baseline, as well as extended versions of PNA and GCNII where value and query/key matrices are not shared.
> - **Table highlights.** We are afraid that we do not fully understand why the reviewer mentions improper highlighting, and thus we would appreciate further clarifications.
> To further clarify our approach, we highlight whether (L-)CAT improves over its baseline GAT model (bold), *as well as the method with the best average performance (underlined).* This is stated in the table captions, and we followed this rule to avoid visual cluttering, but we are open to suggestions on how to better present the results.

---

### Official Review · Reviewer_kUTn · 2022-10-26

**Confidence:** 3
**Correctness:** 3
**Technical Novelty And Significance:** 3
**Empirical Novelty And Significance:** Not applicable
**Recommendation:** 6

**Clarity, Quality, Novelty And Reproducibility:**

**************Clarity**************

The paper is clearly written to understand overall method.

**************Quality**************

Some experiments for baselines are not properly conducted.

**********Novelty**********

The proposed method has a fair novelty.

******************************Reproducibility******************************

The paper has good reproducibility.

**Strength And Weaknesses:**

**Strengths**

(1) This paper shows the effectiveness of proposed methods with theoretical analysis.

(2) The proposed CAT shows consistently good performance across datasets.

(3) Experiments on synthetic datasets prove the CAT’s effectiveness compared to GCNs and GATs.

**Weakness**

(1) The formulation GCNs is different from the formulation proposed in [1] on undirected graphs. Since the datasets used in experiments are undirected graph, it is more appropriate to use $\tilde{D}^{-1/2}\tilde{A}\tilde{D}^{-1/2}$.

************************Additional Questions************************

(1) I wonder the performance of GCN, GAT, and CAT when using more than 1 layer on the synthetic dataset.

---

[1] Fountoulakis, Kimon, et al. "Graph Attention Retrospective." arXiv 2022.

**Summary Of The Paper:**

The paper proposes graph Convolutional Attention Networks (CAT) to exploit the advantages of Graph Convolutional Networks (GCNs) and Graph Attention Networks (GATs). Further, the authors design Learnable graph Convolutional Attention Networks (L-CAT), which softly selects GCNs, GATs, and CAT. The paper presents theoretical analysis of CAT and conduct node classification tasks using ten datasets.

**Summary Of The Review:**

Overall, I think this paper is marginally over the acceptance threshold. The paper demonstrates the effectiveness of their methods through theoretical analysis.

---

> ### Author Response · Authors · 2022-11-09
> **Reply to reviewer kUTn**
>
> Many thanks for the encouraging feedback, below we clarify the individual items raised in the review.
> - **GCN formulation.** We fail to see how our formulation differs from that of [[1] Kimon et. al.](https://arxiv.org/abs/2202.13060), as they adopt the ansatz of [[2] Baranwal et. al.](https://arxiv.org/abs/2102.06966), which uses the same formulation as we do in Eq. (2) (see section 1.3 of the latter paper). However, we do agree that our GCN formulation is simple and that other variants can perform better in terms of downstream-task performance. For that reason, we describe in Appendix F (which we have expanded with new experiments) how to extend L-CAT to other GNN models.
> - **Synthetic dataset with multi-layer GNNs.** The synthetic dataset, while simple, exploits some key properties (such as linear separability) that let us clearly analyze the expressiveness of a GNN with a single layer. We do not expect the synthetic task to be hard to solve in the multi-layer setting, and more complex datasets would be needed to challenge such models. Indeed, [[3] Baranwal et. al.](https://arxiv.org/abs/2204.09297) recently showed that a GCN layer + MLP is sufficient to separate a similar non-linearly separable dataset. We point this out in footnote 2 of page 3.
> - **Experiments quality.** We would appreciate if the reviewer could be more specific regarding their expectations on our experiments, so we have addressable items during the rebuttal period.
>
> [1] Fountoulakis, Kimon, et al. "Graph Attention Retrospective." *arXiv preprint arXiv:2202.13060 (2022).*
>
> [2] Baranwal, Aseem, Kimon Fountoulakis, and Aukosh Jagannath. "Graph convolution for semi-supervised classification: Improved linear separability and out-of-distribution generalization." *arXiv preprint arXiv:2102.06966 (2021).*
>
> [3] Baranwal, Aseem, Kimon Fountoulakis, and Aukosh Jagannath. "Effects of Graph Convolutions in Deep Networks." *arXiv preprint arXiv:2204.09297 (2022).*

---

> > ### Comment · Reviewer_kUTn · 2022-11-25
> > **Thank you for your response.**
> >
> > Thank you for your response on my review.
> >
> >  For the experiments quality, I wanted to see the result of the proposed L-CAT with graph convolutional networks ($\tilde D^{-1/2} \tilde A \tilde D^{-1/2}$ and authors described what I wanted in Appendix F of revised version.
> >
> > There are no further questions from my side at this point.

---

### Official Review · Reviewer_UgAb · 2022-10-27

**Confidence:** 4
**Correctness:** 4
**Technical Novelty And Significance:** 3
**Empirical Novelty And Significance:** 3
**Recommendation:** 8

**Clarity, Quality, Novelty And Reproducibility:**

As described in the above section, the motivation and the problem definition are clear.

Their approach is logical and seems to have sufficient applicability.

**Strength And Weaknesses:**

- The overall quality of the writing and organization of the work is quite high.

- The background of the study and the significance of the contribution are laid out in great detail.

- The second and third parts (preliminaries and limitations of GCN and GAT) offer an insightful summary of GNNs.

- Findings from theory are addressed, and they are also confirmed by empirical research.

- In general, improving graph neural networks by interpolating across different layer types using parameters that can be learned in an end-to-end manner seems promising.


**Summary Of The Paper:**

Combining Graph Convolutional Networks (GCNs) with Graph Attention Trees (GATs), the authors propose a novel architecture for Graph Neural Networks, Learnable Convolutional Attention Layer (L-CAT).


In order to bridge the gap between the graph convolutional layer, the graph attention layer, and the graph convolutional attention layer, this paper introduces a new layer type for graph neural networks.


They add two scalar parameters per layer and learn them along with the other parameters in an end-to-end fashion. Inspiring theoretical conclusions based on prior work are presented, and the authors' suggested strategy is experimentally validated.

**Summary Of The Review:**

I recommend to accept the paper.

---

> ### Author Response · Authors · 2022-11-09
> **Reply to reviewer UgAb**
>
> We thank the reviewer for all the positive feedback. Hearing that our work is insightful, promising, clear, and of high quality, is nothing less than flattering and reassuring. We share the enthusiasm and believe that L-CAT enables not only more robust results, as demonstrated by our experiments, but also offers a way of readily incorporating layer selection into the training process, reducing the need for cross-validation.

---

### Author Response · Authors · 2022-11-09
**Generaly reply**

We thank the reviewing team for their effort and insightful feedback, which will ultimately improve the state of the manuscript. We appreciate the kind words from the reviewers, as well as the unanimous content regarding the problem motivation, theoretical results and synthetic experiments. We address individual questions as replies to the respective reviews.

As part of the rebuttal, we have updated the manuscript with the following two changes:
1. We have introduced an additional paragraph at the end of section 5, summarizing appendix F and better referring the reader to it, as it explains how to extend L-CAT to GNN models other than the GCN introduced in Eq. 2 (e.g., the symmetric normalized GCN, PNA, or GCNII).
2. We have expanded the experimental section in Appendix F.1, which previously showed results combining L-CAT with the PNA model, to also include results of an extended version of PNA enabled by our framework (where value matrices are different of key/query matrices), as well as results on combining L-CAT with the GCNII model.

---

### Decision · Program_Chairs · 2023-01-20

**Decision:**

Accept: poster

**Justification For Why Not Higher Score:**

The approach is nice and has significant merit to be accepted.
However, the approach is a simple (albeit noteworthy) extension of existing approaches.

**Justification For Why Not Lower Score:**

The theoretical justification of l-CAT is rather elegant. The evaluation of the method on various benchmarks clearly highlight its merit.

**Metareview: Summary, Strengths And Weaknesses:**

__Summary.__ This paper investigates efficient aggregation/attention layers in GNN architectures. More specifically, to go beyond the usual dichotomy between aggregation (GCNs) and attention (GAT), they begin by introducing a  new type of layer, CAT (convolution attention layer), which computes an attention score after the usual aggregation step. They show the performance of this new type of "pooling" layer, showing its benefits on several datasets and highlight its theoretical competitive advantage in a Contextual SBM model. To forego the need to then "tune" the architecture  and select the "right" pooling mechanism, the authors propose to use a learnable convolution attention network that interpolates between GCN, GAT and their  layer, CAT. The weights ($\lambda_1$ and $\lambda_2$) are learned jointly with the parameters of the networks. They show that such a layer provides better performance of the GNN under higher levels of edge and feature noise.


__Feedback from the reviewers.__ Overall, the reviewers seem to agree that this is a good paper and that the approach has merit. The proposed mechanism seems to work, as highlighted by the extensive experiments proposed by the authors. The paper is nicely written --- with, we note, a nice theoretical justification of their proposed layer---, and the experiments are very complete (nice investigation of the robustness). Initial reviews raised objections regarding the limited evaluation methods (the authors had just used GCN as an architecture), but the authors have extended their evaluation and used l-CAT with various type of GNN architectures during the rebuttal period.  The contribution is therefore significant enough to warrant admission.


**Note From Pc:**

if the above contains the word "oral" or "spotlight" please see: "oral" presentation means -> notable-top-5% and "spotlight" means -> notable-top-25%. As stated in our emails, we are disassociating presentation type from AC recommendations

**Summary Of Ac-Reviewer Meeting:**

N/A